

# Spatially continuous snow depth mapping by airplane photogrammetry for annual peak of winter from 2017 to 2021

Leon J. Bührle[1,2], Mauro Marty[3], Lucie A. Eberhard[1,2,4], Andreas Stoffel[1,2], Elisabeth D. Hafner[1,2,5], Yves Bühler[1,2]

[1] WSL Institute for Snow and Avalanche Research SLF, Davos Dorf, 7260, Switzerland
[2] Climate Change, Extremes and Natural Hazards in Alpine Regions Research Center CERC, 7260 Davos Dorf, Switzerland
[3] Swiss Federal Institute for Forest, Snow and Landscape Research WSL, Birmensdorf, 8903, Switzerland
[4] Institute of Geodesy and Photogrammetry, ETH Zurich, Zurich, 8092, Switzerland
[5] EcoVision Lab, Photogrammetry and Remote Sensing, ETH Zurich, Zurich, 8092, Switzerland

*Correspondence to*: Yves Bühler (buehler@slf.ch)

**Abstract.**

Information on snow depth and its spatial distribution is important for numerous applications such as the assessment of natural hazards, the determination of the available snow water equivalent (SWE) for hydropower, the dispersion and evolution of
flora and fauna and the validation of snow-hydrological models. Due to the heterogeneity and complexity of snow depth distribution in alpine terrain, only specific remote sensing tools are able to accurately map the present variability. To cover large areas (>100 km²), airborne laser scanners (ALS) or survey cameras mounted on piloted aircrafts are needed. Applying the active ALS leads to considerably higher costs compared to photogrammetry but also works in forested terrain. The passive photogrammetric method is more economic, but limited due to its dependency on good acquisition conditions (weather,
sufficient light, contrast on the snow surface). In this study, we demonstrate the reliable and accurate photogrammetric processing of high spatial resolution (0.5 m) annual snow depth maps during peak of winter over a 5-year period under different acquisition conditions within a study area around Davos, Switzerland. Compared to previously carried out studies, using the new Vexcel Ultracam Eagle M3 survey sensor, improves the average ground sampling distance (GSD) to 0.1 m at similar flight altitudes above ground. This allows for very detailed snow depth maps, calculated by subtracting a snow-free
digital terrain model (DTM acquired with ALS) from the snow-on digital surface models (DSMs) processed from the airborne imagery. Despite complex acquisition conditions during the recording of the Ultracam images (clouds, shaded areas and new-snow cover), 99 % of unforested areas were successfully reconstructed. We applied masks (high vegetation, settlements, waters, glaciers) to significantly increase the reliability of the snow depths measurements. An extensive accuracy assessment including the use of check points, the comparison to DSMs derived from unpiloted aerial systems (UAS), and the comparison
of snow-free pixels to the ALS-DTM prove the high quality and accuracy of the generated snow depths. We achieve a root mean square error (RMSE) of approximately 0.25 m for the Ultracam X and 0.15 m for the successor sensor Ultracam Eagle M3. By developing an almost automated, consistent and reliable photogrammetric workflow for accurate snow depth



distribution mapping over large regions, we provide a new tool for analysing snow in complex terrain. This enables more detailed investigations on seasonal snow dynamics and serves as ground reference for new modelling approaches as well as

satellite-based snow depth mapping.

# 1 Introduction

Accurate snow depth mapping is important for the assessment, prediction and prevention of natural hazards such as snow avalanches or floods. For example, crack propagation and the size of release areas of snow avalanches are linked to snow depth distribution (Schweizer et al., 2003; Veitinger and Sovilla, 2016). The release volume and the length of avalanche

runouts particularly depend on the releasing and eroded snow volumes. Snow-drift caused by wind and its influence on the snow distribution in the starting zone and the avalanche path is another key parameter for avalanches (Schön et al., 2015). Further hazards related to snow depths are snow loads on buildings, threatening not only the stability of roofs but potentially leading to dangerous snow slides from roofs (Croce et al., 2018). Additionally, snow depth and the snow water equivalent (SWE) of the snowpack are crucial for the forecast of floods during snowmelt periods. Various economic sectors can benefit

from accurate snow depth information. For example, the available SWE within catchments is a key criterion for the implementation and operation of hydroelectric power plants (Magnusson et al., 2020). Detailed information on snow depth distribution on slopes is also valuable for the decision-makers of winter resorts (Spandre et al., 2017). Moreover, snow depth mapping facilitates research on interactions between snow depth distribution and flora / fauna (Wipf et al., 2009), because snow depth distribution determines the growth of plants and shapes the habitats of mountain animals.


Precise snow depth measurements are key data when developing and validating models for snow parameters. For avalanche modelling tools such as the Rapid Mass Movement Simulation (RAMMS) (Christen et al., 2010) or SAMOS AT (Sampl and Zwinger, 2004), the snow volume in the starting zone as well as in the deposition zone can be compared to modelled results. Furthermore, they can serve as reference datasets for snow depth distribution models (Wulf et al., 2020) and snow-

hydrological models like Alpine3D (Richter et al., 2021; Schlögl et al., 2018), Factorial snowpack model (FSM) (Essery, 2015) and Crocus (Brun et al., 1992). Snow depth maps can also improve crucial input data for snow hydrological models like precipitation by serving as precipitation scaling parameter (Brauchli et al., 2017; Vögeli et al., 2016). From high spatial resolution snow depth maps, the derivation of the standard deviation of snow depths during peak of winter is feasible, and can be used for various model applications (Helbig et al., 2021).


Traditionally, snow depth is measured by field observations such as manual probing or by automated weather stations (AWS). However, to get spatially continuous coverage, interpolation is required. As snow depths vary a lot over short distances, especially in complex terrain (Grünewald et al., 2010; Grünewald et al., 2014), interpolation is insufficient for most applications and has been shown to result in very large errors. Ground-penetrating radar (GPR) can capture many point



measurements when mounted on a sledge or snowmobile (Helfricht et al., 2014) with a high accuracy of less than 0.1 m
      (Griessinger et al., 2018). Still, transects and not spatially continuous snow depth distribution are measured (McGrath et al.,
      2019). In addition, very steep and poorly accessible terrain cannot be covered and the method is only practicable for small
      catchments, if no helicopter is available (Prokop et al., 2008).

Remote sensing tools can provide accurate and spatially continuous snow depth measurements. Terrestrial laser scanning
      (TLS), based on the reflectance of laser beams on object surfaces, can provide very exact measurements (Deems et al., 2013).
      The achieved accuracy depends on the sensor and the object's distance from the scanner and is ranging from 0.05 m to 0.2 m
      in distances below 1000 m (Grünewald et al., 2010; Prokop, 2008) and 0.3 m to 0.6 m over longer distances of around 3000
      m (López-Moreno et al., 2017). Other crucial advantages are a lower weather-dependency regarding the illumination
conditions and possible measurements in vegetated areas, if suitable scanner locations with a sufficient measurement angle
      are available (Revuelto et al., 2015). Limitations of this procedure are the access to locations for the scanner and the occurrence
      of concealed areas which cannot be measured as well as poor weather conditions such as strong snowfall or fog (Prokop,
      2008).

In recent years, the importance of digital photogrammetric methods has increased mainly due to the development of the SIFT-
      algorithm (Lowe, 2004), easy to apply software like Agisoft Metashape or Pix4D and the development of unpiloted aerial
      systems (UAS). Various publications have confirmed the high accuracy of snow depths derived from UAS photogrammetry.
      The accuracy mainly depends on the sensor and the ground sampling distances (GSD), ranging from 0.05 m to 0.2 m (Bühler
      et al., 2016; Harder et al., 2016; Michele et al., 2016). Critical issues for this method are the dependency on good weather and
light conditions (Bühler et al., 2017; Gindraux et al., 2017) and difficulties of measuring snow depths in areas with high
      vegetation. Unpiloted aerial laser scanning systems (ULS) combine the advantages of TLS and UAS and can measure snow
      depths with a high accuracy of around 0.1 m in unforested (Jacobs et al., 2021) as well as 0.2 m in forested terrain (Harder et
      al., 2020). However, the current UAS, ULS and TLS can only capture areas up to 5 km² (Revuelto et al., 2021). To map larger
      regions, airborne laser scanners (ALS), optical cameras mounted on piloted airplanes or satellites are needed. The increased
spatial resolution of imagery from optical satellites such as Pléiades and Worldview-3, combined with their high temporal
      resolution and availability of near-infrared (NIR) bands enable new possibilities for large-scale snow depth mapping. First
      studies have shown that snow depth measurements from Pléiades imagery in comparison to reference data exhibit deviations
      of more than 0.5 m. For example Marti et al. (2016) calculated a normalized median absolute deviation (NMAD) value of
      0.78 m compared to snow depth measurements derived from UAS. Other studies researching the application of Pléiades
achieved a root mean square error (RMSE) relative to reference data of 0.5 m (Eberhard et al., 2021; Shaw et al., 2020) to 0.8
      m (Deschamps-Berger et al., 2020). These accuracies do not satisfy the requirements for most snow depth mapping
      applications. The study of McGrath et al. (2019) applied the WorldView-3 satellite with a spatial resolution of 0.3 m and
      achieved a considerably higher accuracy with a RMSE of 0.24 m compared to GPR. The dependency of optical satellite



sensors on cloud-free conditions stimulates the application of active satellite sensors with synthetic aperture radar (SAR), which are independent from light and weather conditions (Tsai et al., 2019). However, their spatial resolution is currently not sufficient for detailed and accurate snow depth measurements (Lievens et al., 2019; Lievens et al., 2022).

In contrast to satellite measurements, ALS fulfils the criteria for accurate snow depth mapping over large areas. Different studies have proven, that the accuracy, which depends on the flight altitude (Deems et al., 2013), is similar to the one of TLS (Mazzotti et al., 2019). However, Bühler et al. (2015) estimated the cost for an ALS-flight and the processing of the data to around 50'000 to 80'000 CHF, covering an area of 150 km². These high costs prevent the realisation for many implementations. Airplane-based photogrammetry, however, is more economic with costs ranging from 30'000 to 60'000 CHF (Bühler et al., 2015). Despite the application of a lower cost camera (Nikon D800E), Nolan et al. (2015) reached an excellent accuracy of about 0.1 to 0.2 m. Bühler et al. (2015) also produced a high-resolution snow depth map with a spatial resolution of 2 m, covering a heterogeneous high-mountain area of 300 km² around Davos. Using the surveying pushbroom-scanner Leica ADS 80, a RMSE of 0.3 m comparing to GPR, TLS and manual probing was achieved. Meyer et al. (2021) created snow depth maps with a 1 m spatial resolution and demonstrated that airplane-based photogrammetry can reach accuracies similar to the ones of the ALS. The new state of the art 450 Megapixel (MP) frame sensor Vexcel Ultracam Eagle M3 can record extremely high spatial resolution images, which enables the generation of accurate digital surface models (DSM) of large regions. Eberhard et al. (2021) achieved an accuracy of around 0.1 m using the Vexcel Ultracam Eagle M3 as well as 29 ground control points (GCPs) to refine the orientation within a small catchment (30 km²).

In this study, we present the consistent processing of five annual snow depth maps with a spatial resolution of 0.5 m based on Vexcel Ultracam images covering approximately 250 km². These datasets were acquired at peak of winter, which was characterized by large differences in snow-cover and average snow depth, under various weather and illumination conditions. Complementing the existing snow depth maps from 2010 to 2016 (2 m spatial resolution) (Bühler et al., 2015; Marty et al., 2019) these unique data series enable the long-term monitoring of snow depth distribution in high alpine terrain.

## 2    Study area Davos, Switzerland

The study area is located around Davos in eastern Switzerland. Varying flight routes during image acquisition resulted in different extents of the derived snow depth maps. However, the core area with an extent of 240 km² was covered by all flight campaigns from 2019 to 2021 and will also serve as study area for the prospective snow depth maps. Various regions of the research area, for example, the Weissfluhjoch (Wirz et al., 2011), the Wannengrat (Schirmer and Lehning, 2011), the Dischma Valley (Bühler et al., 2015; Eberhard et al., 2021; Grünewald et al., 2014), and the Flüela Valley (Bühler et al., 2016) were already part of snow depth mapping studies (Fig. 1). The diversity of the terrain, including settlements, heavily forested and glaciated areas as well as extremely steep faces, is representative for many mountain regions. The elevation of the main study area is ranging from 1100 m a.s.l around Klosters to the 3229 m a.s.l. high Piz Vadret. The research area is located in a transition zone between the humid north-alpine climate and the climate zone of the central alps, which is significantly drier





(Kulakowski et al., 2011; Mietkiewicz et al., 2017). The main snowfall in the winter season is recorded during north-westerly and northern weather situations, which are commonly connected to strong storms with high wind speeds (Gerber et al., 2019; Mott et al., 2010). The date of the peak of winter in this region is usually between March and April at an elevation of approximately 2000 m a.s.l.. Previous investigations have shown, that the average snow depth of the study area varies a lot and can range from around 1 m to 2.5 m (Marty et al., 2019).

**Figure 1.** Overview of the study area: Snow depth map generated by the airplane 2017 (black), extent of snow depth map from 2018 (blue) and snow depth area derived from the respective flights in 2019, 2020 and 2021 (red; corresponds to main study area). Additionally, the area covered by the UAS for reference data in 2018 and 2021 are shown (green). The red polygon in the inset map depicts the location of the main study area in Switzerland (map source: Federal Office of Topography).



# 3    Data and sensor

## 3.1    Vexcel Ultracam

Airborne imagery was acquired with the survey camera Vexcel Ultracam series. The Vexcel Ultracam X was applied in 2017, and is characterized by a sensor pixel size of 7.2 µm x 7.2 µm, a focal length of 100.5 mm and a resolution of 9420 x 14430

pixels (Schneider and Gruber, 2008). Due to better characteristics of the successor camera Ultracam Eagle M3, the Ultracam X was replaced in the following years. The Ultracam Eagle M3 belongs to the current state-of-the art cameras for photogrammetric measurements with 450 MP (Bühler et al., 2021). The improvements include a better sensor pixel size of 4 µm x 4 µm, a focal length of 120.7 mm and a better resolution of 26000 x 14000 pixels (Eberhard et al., 2021). Both Ultracam cameras simultaneously acquire the four spectral bands red-green-blue (RGB) and NIR with a radiometric resolution of 14

bits. The camera positions are registered by differential global navigation satellite system (DGNSS) with a nominal accuracy of 0.2 m. The orientation of the camera is recorded through an inertial measurement unit (IMU) with a nominal accuracy of 0.01° (omega, phi, kappa). This data simplifies the determination of interior orientation and the correct georeferencing and prevents tilts of the DSM.

The flights were conducted during the expected peak of winter between March and April around midday to avoid large, shaded

areas. The exact extent of each flight varied from year to year and is based on the authorized flight route, weather conditions and occurrence of technical problems (Fig. 1). The captured region in 2017 covered 600 km² (Fig. 1, black polygon) and was considerably larger than in the following years. High costs and limited flight permissions resulted in the selection of a smaller main study area (250 km², red polygon) around Davos for the subsequent years.

Simultaneously to the flights, reference points were marked with specially mustered tarps and measured by DGNSS with a

vertical accuracy of 0.05 m. Because no reference points were acquired in 2017 and 2019, ten extra reference points on conspicuous road markings were measured in retrospect. Different characteristics and special features of each flight are described in the Table 1.





**Table 1.** Properties of the executed annual Ultracam flights during peak of winter.

| Acquisition date | Sensor type | Reference points | Mean GSD [m] | Mean flight altitude [m above ground] | Notice |
|---|---|---|---|---|---|
| 16 March 2017 | Ultracam X | 0 | 0.23 | 3430 | Large, shaded areas, inaccuracies of NIR-band |
| 11 April 2018 | Ultracam Eagle M3 | 8 | 0.06 | 1780 | Technical problems (airplane), heavily cloudy |
| 16 March 2019 | Ultracam Eagle M3 | 0 | 0.12 | 4040 | Only RGB-bands, no NIR-band |
| 6 April 2020 | Ultracam Eagle M3 | 38 | 0.12 | 3970 | Good conditions |
| 16 April 2021 | Ultracam Eagle M3 | 14 | 0.12 | 3910 | few clouds in the east and west part, new snow |

## 3.2    Reference dataset

### 3.2.1    Airborne laser scanner (ALS) from summer 2020

Calculating snow depths with photogrammetric methods requires an accurate snow-free reference dataset. For the main part of the study area, an ALS point cloud from summer 2020 was available (Federal Office of Topography swisstopo, 2021a).

The specified accuracies of 0.2 m in horizontal and 0.1 m in vertical direction comply with the requirements of accurate snow depth mapping. The point density of the ALS point cloud of at least 5 points m$^{-2}$ and on average 20 points m$^{-2}$ allows the exact reconstruction of small-scale features and extremely steep faces. The exact point classification enables the separation of vegetation, ground, buildings and waters. Correspondingly, a digital terrain model (DTM), a normalized ALS-DSM which only considered vegetation and a normalized-DSM which only took buildings into account were processed. Due to the high

point density a rasterization to a spatial resolution of 0.5 m was applied. The ALS-DTM also served as reference dataset to evaluate the accuracy of the snow depth maps through the comparison of snow-free areas.

### 3.2.2    Unpiloted aerial systems (UAS) photogrammetry 2018 and 2021

To compare spatial snow depths of small catchments, UAS-derived DSMs are commonly used, given the vertical accuracy of often better than 0.1 m (Bühler et al., 2016). Therefore, two UAS flights were carried out for a small subset (3.5 km²) in the

Dischma Valley during the Ultracam flight campaigns in 2018 and 2021 (Fig. 1, green polygon). Short-term technical problems on the airplane in 2018 prevented the simultaneous capture of the UAS (eBee+ RTK) and airborne data. No significant snowfall, but slightly positive temperatures were registered between the recording of the UAS and the airborne





data. In 2021, the UAS acquisition was conducted simultaneously to the Ultracam flight by the vertical take-off and landing (VTOL) drone WingtraOne with a 42 MP camera. The processing workflow of the UAS-derived DSMs was similar to the described approach in Eberhard et al. (2021) with the crucial difference that outliers of the point cloud were identified and excluded. Outliers were removed by the confidence interval of less than 3 depth maps. The achieved accuracy of the DSMs was identified by using check points (CP) with a RMSE of 0.06 m (2021) and 0.1 m (2018).

### 3.2.3 Manual reference points

The manually measured reference points during the Ultracam flights had the purpose of serving as GCPs or CPs. Due to the time-consuming and expensive fieldwork as well as the presenting avalanche danger, the number of reference points was limited and often close to streets. The ten points measured in retrospect were valid as reference, although they have a lower reliability compared to direct measured reference points.

Only in 2020, during low-level avalanche danger, 40 reference points could be placed well distributed over different elevations and on various aspects.



## 4 Methods

The creation of reliable and accurate snow depth maps consists of four steps (Fig. 2):

- Processing of airborne imagery and ALS point cloud
- Calculation of snow depths
- Creation and application of necessary masks
- Accuracy check of the finalized snow depth maps

To prevent offsets, using the same coordinate system for all
processing steps is fundamental. The horizontal coordinate system CH1903+ LV95 and the height reference system LN02 were selected due to their common application in Switzerland. Conversions from other coordinate systems were carried out with the tool REFRAME (Federal Office
of Topography swisstopo, 2021b) and transformations available in ArcGIS Pro 2.7. The processing of airborne imagery was realized in Agisoft Metashape 1.6, an easy-to-use software for photogrammetric applications. Using newer versions of Agisoft Metashape such as 1.7 resulted in
the occurrence of numerous holes in the DSM, given that the formation of the dense cloud in 1.7 is characterized by a more aggressive filtering. Nevertheless, Agisoft Metashape has proven its value in numerous snow-related studies (Avanzi et al., 2018; Bühler et al., 2016) and allows the
processing of very large high spatial resolution airborne images (Eberhard et al., 2021; Meyer et al., 2021). The calculations and modifications of the snow depth values were realized in ArcGis Pro. Handling numerous very high-resolution images requires a high-end hardware setup. We
used an Intel® Xeon Platinum 8267 2.9 GHz processor (48 cores), 383 GB of memory and six NIVIDIA GeForce RTX 2080 Tì`s with 12 GB of graphic memory. The processing time of each Ultracam dataset was approximately two days.

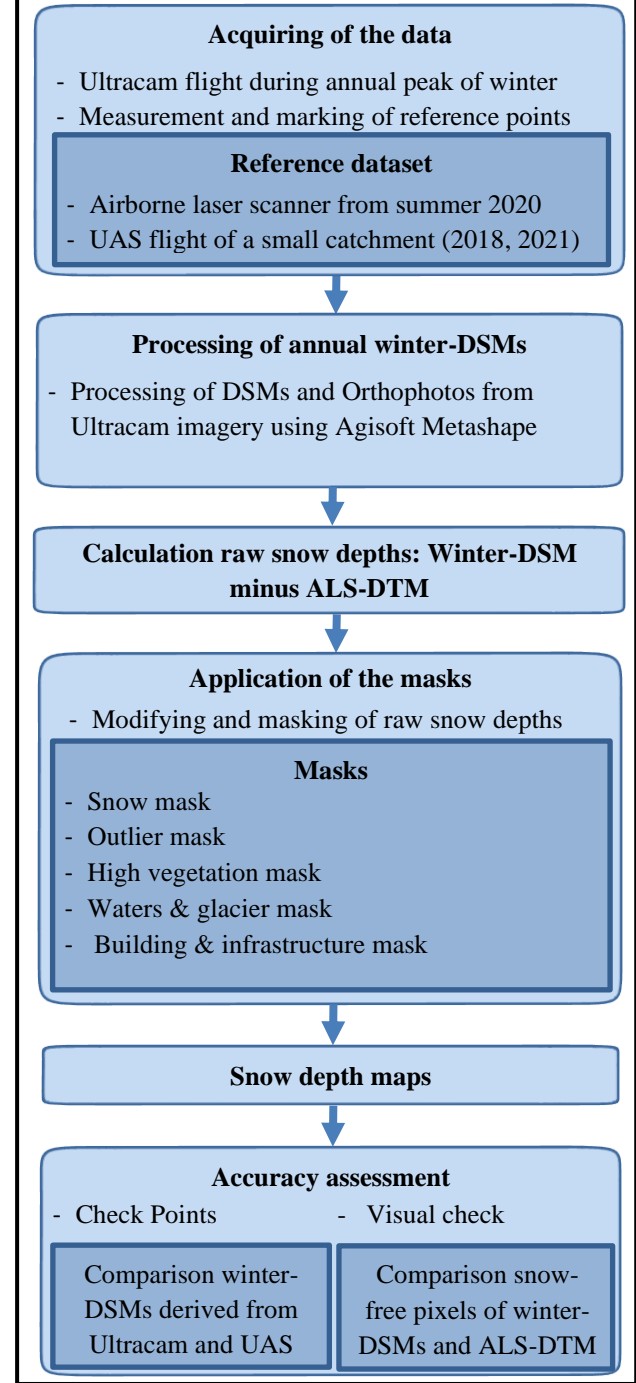

**Figure 2.** Flowchart illustrating all processing steps for the creation of snow depth maps.



## 4.1    Processing workflow of airborne imagery

The processing of aerial images in Agisoft Metashape is based on the Structure from Motion (SFM) algorithm (Koenderink and van Doorn, 1991; Westoby et al., 2012). The general workflow is well-explained in various publications (Adams et al., 2018) and in the Agisoft manual (Agisoft LLC, 2020). However, the existing framework conditions of this study, applying the new high-

resolution sensor Vexcel Ultracam in combination with such a large and heterogeneous study area are sparsely explored. Only Eberhard et al. (2021) successfully generated a winter-DSM derived from Ultracam imagery. Therefore, the workflow used in this study is based on Eberhard et al. (2021). However, the aim to use as few as possible GCPs to refine the orientation due

to the limited availability of reference points required an adaption of this workflow.

The Vexcel Ultracam camera is regularly calibrated, hence the interior orientation is known exactly. However, the application of the calibrated lens distortion parameters led to a large offset of the z-value in the resulting DSM of

approximately 2 m. Therefore, the interior and exterior orientation were calculated in Agisoft Metashape during the bundle-adjustment process (Triggs et al., 2000). Subsequently, the parameters for the interior and exterior orientations (especially focal length) were improved by the application of two to five GCPs. The necessary number of GCPs and the influence of their

distribution for an exact and reliable orientation was determined by a parameter study for the Ultracam flight in 2020, which was characterized by 40 well-distributed reference points. This approach has shown that the use of only one GCP is sufficient for a correct orientation and determination of atmospheric distortions under favourable acquisition conditions. Warps and tilts, which often

occur using a low number of GCPs with a limited dispersion over the area, were avoided because of the availability of the exact coordinates of the shutter release points and the rotation angles of the camera. Using more and well distributed GCPs had no significant influence on the quality grade (Table 2). However, due to the high dependence on the precise measurement when using only one GCP,

and the possibility of varying atmospheric distortions when using cloud-covered images, the implementation of two to five GCPs is to be preferred.

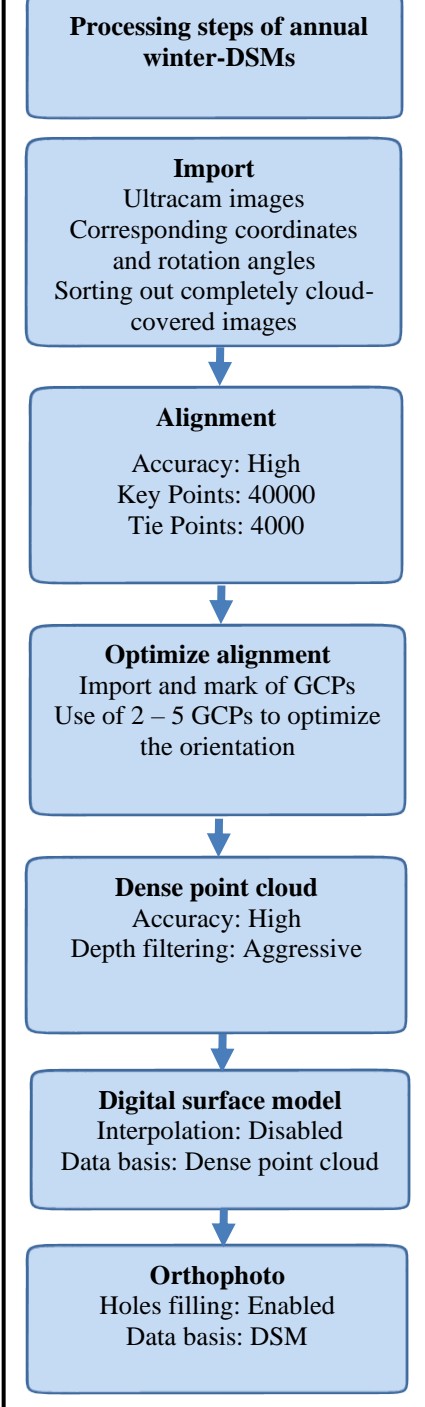

**Figure 3.** Flowchart illustrating the workflow generating the winter-DSMs on the basis of Ultracam images.



**Table 2.** Overview settings used and the corresponding accuracy (RMSE) for the parameter study for 2020.

| Pre-calibration | Used dGNSS coordinates | Used rotation angles | Number and distribution GCPs | RMSE (Z; Total) [m] |
|---|---|---|---|---|
| ✓ | ✓ | ✓ | - | 2.19; 2.20 |
| ✗ | ✓ | ✓ | - | 1.04; 1.08 |
| ✗ | ✓ | ✓ | 15 | 0.07; 0.15 |
| ✗ | ✓ | ✓ | 8 | 0.08; 0.17 |
| ✗ | ✓ | ✓ | 4 (Davos, Dischma, Sertig) | 0.07; 0.155 |
| ✗ | ✓ | ✓ | 3 (Davos, Dischma, Frauenkirch) | 0.08; 0.16 |
| ✗ | ✓ | ✓ | 1 (Davos) | 0.08; 0.17 |

The final DSMs and the corresponding orthophotos were exported with a spatial resolution ranging from 0.1 to 0.45 m,
depending on the average GSD. The workflow used is illustrated in Fig. 3 and further settings are described in Eberhard et al.
(2021).

### 4.2 Creation of snow depth maps

The snow depths were calculated by subtracting the photogrammetric winter-DSM from the ALS-DTM. The improved spatial
resolution of the Ultracam-based DSMs and the ALS-DTM (0.5 m) enables the creation of snow depth maps with a spatial
resolution of 0.5 m. Therefore, the winter-DSMs were resampled to a resolution of 0.5 m. Due to low vegetation is often
compressed by snow in winter (Feistl et al., 2014), the application of an ALS-DTM was preferred over an ALS-DSM, as the
use of a DSM would underestimate the snow depths in regions with low vegetation (Eberhard et al., 2021). Nevertheless,
using a DTM leads to extremely high and unrealistic snow depths in settlements and forested areas, which both have a high
influence on various statistics of the snow depths. Consequently, these areas exhibit many inaccuracies of photogrammetric
measurements and they are not suitable for accurate snow depth mapping. Furthermore, different characterizations of high
vegetation regarding its behaviour in the winter months complicates the assessment of the actual snow depth (Eberhard et al.,
2021).



To get accurate and reliable snow depth maps, the application of various masks was required. This procedure is based on the approach of Bühler et al. (2015), improved by Bührle (2021) and contains a snow, an outlier, a high vegetation , a waters and glacier mask as well as an infrastructure and building mask. However, the existing algorithm to calculate the masks was adapted due to the use of the better Ultracam sensor and the availability of an accurate and well-classified ALS point cloud. Without the application of these masks, the average snow depth value of the study area would be overestimated by approximately 1 m. Another goal is the consistent and automatic creation of the masks. Correct snow depth maps depend on the implementation of the masks in the workflow presented (Fig. 4). Excluding regions with heavy cloud-cover and outlying areas led to more reliable snow depth values.

### 4.2.1  Snow Mask

The snow mask has the aim to modify calculated snow depths of snow-free areas to zero (Bühler et al., 2016). Therefore, each pixel of the corresponding orthophoto is classified into the categories snow-covered or snow-free, using the Normalized difference snow index (NDSI) classification (Dozier, 1989; Hall et al., 1995) with a threshold around 0. This approach was applied for the years 2017, 2020 and 2021. Technical issues in 2019 prevented the recording of the NIR-band and accordingly to this no NDSI classification could be processed. In 2018, the NDSI classification falsely classified many artefacts such as snow mixed with soil as snow-free. Therefore, another classification method without the necessity of the NIR-band and a better operation in snow mixed with soil, was required. Since the blue band of snow exhibits higher reflectance than the red and green band (Eker et al., 2019) a threshold of the ratio between the blue and red band was used to determine

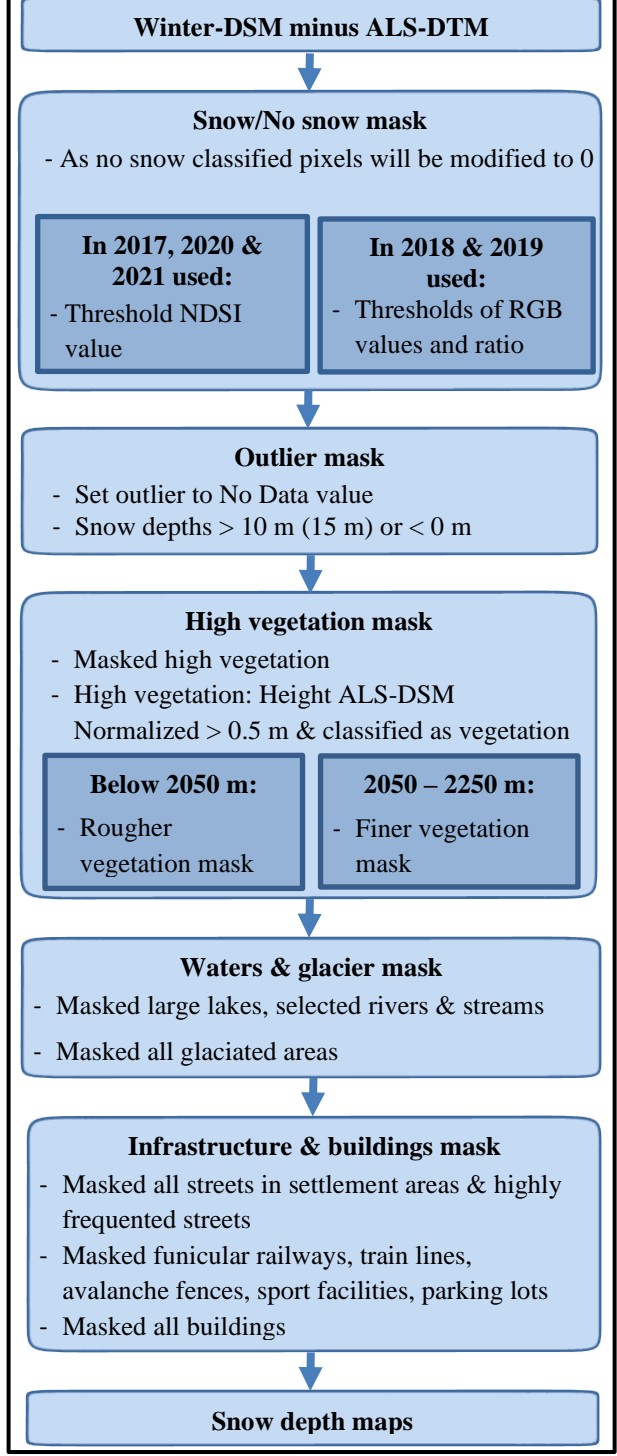

**Figure 4.** Flowchart illustrating the various masks used for the generation of reliable snow depth maps.



the existence of snow. However, the values vary and depend on the strength of shadows, therefore the thresholds were manually determined by an expert. To ensure the reliability of this approach, 500 random points in open terrain in 2019 were selected and manually checked regarding to their correct snow-covered or snow-free classification.

### 4.2.2 Outlier Mask

The outlier mask has the purpose to modify all unrealistic snow depth values, namely negative snow depths and extremely high snow depths above 10 m, to No Data. Only in the year 2019 which was characterized by numerous extremely large avalanche depositions, the upper limit was adapted to 15 m.

### 4.2.3 High vegetation mask

Due to the uncertainties in the actual snow depth and problems with photogrammetric methods around high vegetation, these areas were masked out. Pixels containing high vegetation were identified through the vegetation classification and the calculated object height using the ALS point cloud. High vegetation, defined as vegetation with a height above 0.5 m, only exists up to an elevation of approximately 2250 m in the region of Davos (Rixen et al., 2012). Correspondingly, the calculation of the high vegetation mask was also limited to this elevation. Additionally, a generalization of the high vegetation mask was required because wind, different sensor acquisition characteristics and the various capture times between ALS and Ultracam affected the extent of high vegetation and therefore had an influence on the calculated snow depths. As tree density is dependent on elevation, the algorithm used for the generalization differed between a rougher mask below 2050 m, where dense forests are predominant and a finer mask above 2050 m, where free-standing trees and bushes are prevalent.

### 4.2.4 Waters and glacier mask

The waters and glacier mask prevents unrealistic snow depth values on waters and glaciated areas. The spectral characteristics of water often result in false classification as snow (Durán-Alarcón et al., 2015). Furthermore, due to low water levels during peak of winter, the actual height of snow (HS) on frozen lakes is underestimated. Therefore, larger lakes, rivers and striking streams were masked out. The river courses and shorelines of lakes were obtained from the Swisstopo TLM3D geodata (Federal Office of Topography swisstopo, 2021c).

Another problem occurred on glaciated areas which are characterized by significant thickness losses (approximately 2 m per year in our study area) in the recent years (GLAMOS - Glacier Monitoring Switzerland, 2021). Accordingly, the calculated snow depths from 2017 to 2020 on glaciers also include the loss of ice and overestimate snow depth. Therefore, glaciated areas (stand: 2016) were masked out (Linsbauer et al., 2021).

### 4.2.5 Infrastructure and building mask

The infrastructure and building mask prevents distorted snow depths caused by buildings and by temporary or moveable objects. Therefore, all buildings were completely masked and infrastructure was partly masked. The buildings were derived



from the classified point cloud of the ALS. The locations of technical constructions and infrastructure such as streets were obtained by the Swisstopo database. Railways, funicular railways, sport facilities, parking lots and all streets in settlement areas and main roads outside the settlements were masked out. Technical constructions like avalanche fences were also set to No Data. All masked pixels were generalized by a similar approach as used for the vegetation mask. Within large settlement areas such as Davos, Arosa and Klosters, a very rough generalization and beyond settlements a finer generalization was applied.

### 4.2.6    Masking Overview

Using all presented masks (Fig. 5) on the raw snow depth values resulted in the final snow depth maps. In total, around 67 % of all pixels remained in the snow depth maps. With 28 % the main part of the pixels masked out corresponds to high vegetation areas. With less than 1 %, the number of outliers is considerably lower than outliers observed in Marty et al. (2019), which confirms that this new method is more reliable for the production of snow depths (Table 3).

**Table 3.** Area [km²] and percentage [%] of various masks, outliers and remaining snow depth values for snow depth map 2020

|  | Waters | Glacier | Building & infrastructure | High vegetation | Outlier | Snow depth values |
|---|---|---|---|---|---|---|
| Area [km²] | 1.2 | 2.9 | 6.0 | 67.2 | 1.9 | 160.8 |
| Percentage [%] | 0.5 | 1.2 | 2.5 | 28.0 | 0.8 | 67.0 |







**Figure 5.** Spatial distribution of the masks used in selected extents of the main study area. Darkblue (waters), light blue (glaciers), green (high vegetation) and black (buildings and infrastructure) polygons symbolize the different masks. Rivers are overrepresented for a better identification (map source: Federal Office of Topography).

## 4.3 Accuracy Assessment

An essential part of this study is an extensive accuracy assessment of the snow depth values. Due to the absence of spatially continuous ground truth datasets, we could only determine the accuracy compared to available reference datasets. Basically, only a simultaneously recorded ALS would be able to provide accurate reference data on large scale, this, in turn, would lead to extremely high costs. Therefore, the accuracy assessment consists of five different methods which enabled a conclusive and



spatially continuous evaluation of the annual winter-DSMs. The selected quality procedures for each year depend on the availability and the reliable implementation of reference data (Table 4).

• The first method is to use independent **check points (M1)** (Sanz-Ablanedo et al., 2018). Even though, the number of check points was limited and they were predominantly not well-distributed over the entire study area, they are an

important indicator for the correct orientation of the winter-DSMs.

• Due to their outstanding accuracy, **UAS-derived DSMs (M2)** serve as ground reference and enabled direct and spatial comparison with Ultracam data over a small area

(Deschamps-Berger et al., 2020; Marti et al., 2016).

• **Visual checks (M3)** by an expert examined the plausibility of calculated snow depths and the correct application of the masks over the entire study area.

• Comparisons of snow-free areas of the winter-DSMs

with the reference ALS exhibited further possibilities for evaluations. **Method 4 (M4)** determined deviations on the **main roads** beyond settlements which were always snow-free (Fig. 6). Extreme outliers exceeding 3 m (approximately MBE ± 4 SD) were excluded, because higher deviations were

caused by bridges, tunnels and moveable objects.

M4 was applied in most of the snow depth maps, except 2019. In 2019, the absence of the NIR-band in combination to occurring puddles on the streets resulted in high deviations, which do not correspond to the actual accuracy.

For a significant assessment, streets without puddles were manually digitalized and used as M4 in 2019.

• **Method 5 (M5)** considered deviations of **all other snow-free pixels (M5)** beyond settlements compared to the ALS-DTM. Pixels with vegetation heights exceeding 0.05

**Table 4.** Overview of the accuracy assessment methods performed in the different years.

|  | **2017** | **2018** | **2019** | **2020** | **2021** |
|---|---|---|---|---|---|
| *M1: Check Points* | ✗ | ✓ | (✓) | ✓ | ✓ |
| *M2: UAS* | ✗ | ✓ | ✗ | ✗ | ✓ |
| *M3: Visual check* | ✓ | ✓ | ✓ | ✓ | ✓ |
| *M4: Comparison ALS on streets* | ✓ | ✓ | (✓) | ✓ | ✓ |
| *M5:Comparison ALS beyond streets* | ✓ | ✗ | ✗ | ✓ | ✓ |

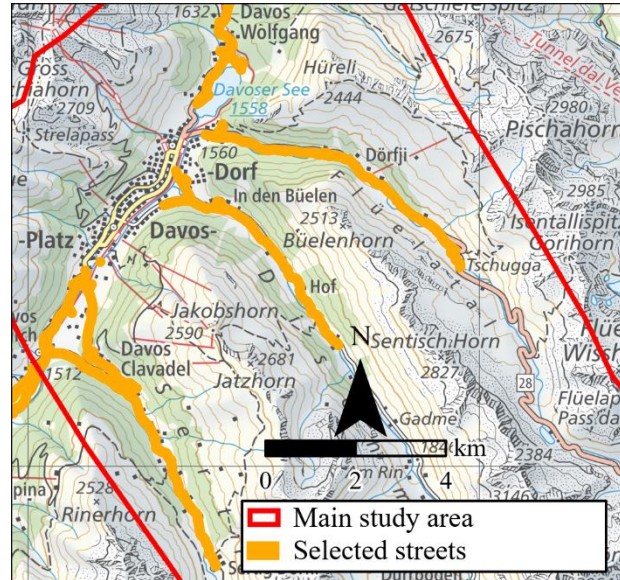

**Figure 6.** Overview roads used (orange lines) for the accuracy assessment method 4, lines are overrepresented for better identification (map source: Federal Office of Topography).

m (derived from normalized ALS-DSM) were excluded. This procedure allowed an accuracy assessment of numerous pixels in the entire study area. The identification of snow-free pixels is based on the snow mask. Despite a high exactness of this method, some falsely classified pixels (e.g snow mixed with soil) appeared in each snow mask. Moreover, the period between the acquisition of the ALS and the winter-DSMs led to occasional changes in the region, such as destroyed forested areas,





newly constructed buildings and existence of temporary objects. Therefore, extreme outliers exceeding 3 m were excluded.

Limited snow-free areas beyond streets in the winters 2018 and 2019 impeded a meaningful evaluation on snow-free pixels in these years.

The quantitative procedures were evaluated by various commonly used statistical quality grades such as mean bias error (MBE), standard deviation (SD), RMSE, median (MdBe) and NMAD (details in Durán-Alarcón et al., 2015; Eberhard et al., 2021 and in Höhle and Höhle, 2009).

The significant impact of a few pixels with high deviations caused by described distortions through temporary objects or false detection of snow-free pixels makes an additional filtering of the deviations necessary. This approach excluded all deviations beyond the confidence interval (MBE ± 2 SD) and was applied for the accuracy assessments of the ALS and the UAS.

Finally, since the accuracy of the snow depth values is also dependent on the exactness of the **reference ALS-DTM**, we have examined the specified accuracy (Z = 0.1 m). Therefore, we compared 24 reference points (see section 3.2.3) on snow-free
areas with the ALS-DTM.

## 5 Results and validation

### 5.1 Accuracy Assessment

The quantitative part (M1, M2, M3, M4) of the accuracy assessment evaluates the deviations of the winter-DSMs to a selected reference dataset. The RMSE value comparing the ALS-DTM to 24 reference points of 0.03 m demonstrates the high
reliability of the reference ALS.

### 5.1.1 2017

The accuracy assessment of the winter-DSM 2017 calculates RMSE values of 0.26 m on open streets and 0.3 m on snow-free pixels after outlier removal
(MBE ± 2SD) (Table 5). Resulting dispersions of method 4 (SD = 0.33 m, NMAD =0.28) and method 5 (SD = 0.42 m, NMAD =0.32) have considerably higher values compared to the other years, which show the lower reliability of this winter-DSM based
on the precursor sensor Vexcel Ultracam X (see section 3.1; Table 1). The same result can be clearly seen at the significantly larger interquartile range of the winter-DSM in 2017 in Fig. 10. Additionally, Fig. 7 shows the difference of

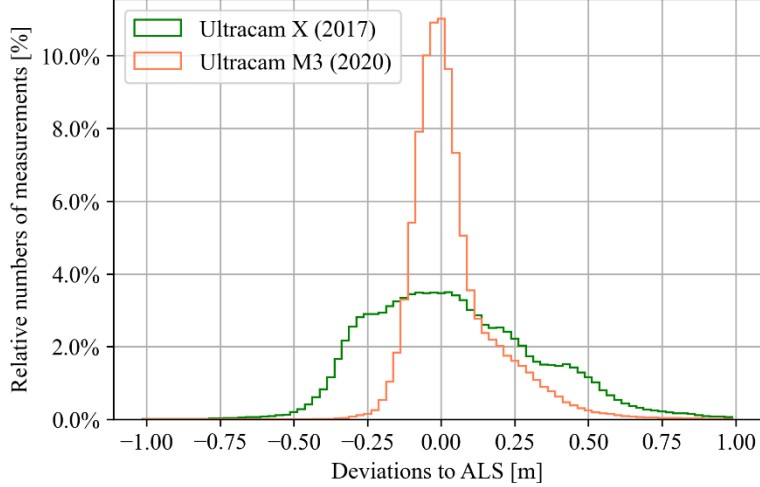

**Figure 7.** Comparison of the dispersion of deviations on streets to ALS between Ultracam X (green, 2017) and Ultracam Eagle M3 (light red, 2020).



the accuracy and corresponding dispersion between Ultracam X and the successor Ultracam Eagle M3. The impact of the higher deviations is evident regarding the high number of outliers (negative snow depths) in complex terrain. Furthermore, inaccuracies of the NIR-band led to insufficient classification of the snow masks. Accordingly, numerous pixels on streets, in transition zones of snow-free to snow-covered and in shaded areas are falsely classified as no snow.

### 5.1.2 2018

In 2018, large cloud-covered areas were excluded from the processing. Therefore, missing images and partly cloud-covered images caused less overlap in these regions and complicated the processing of the remaining study area. Nevertheless, the deviations of the ten check points (RMSE 0.13 m) and the comparison with the UAS-derived DSM (RAW = 0.12, Filtered = 0.09 m) demonstrate a high accuracy of the winter-DSM (Table 5). The aspect-dependency of the deviations between UAS and Ultracam (Fig. 8) can be explained by the four days delay in capture time and therefore compression of the snowpack on

southern slopes due to mild temperatures and strong solar radiation. However, the clouds caused slight regional inaccuracies of the snow depths. The median of method 4 (RAW = 0.08, Filtered = 0.09) shows a slight overestimation of the snow depths, which especially occurred in the south of

Davos close to cloud-covered areas. Despite these overestimations, the RMSE (RAW = 0.18, Filtered = 0.16) of the deviations on roads also proves the spatially high reliability of the winter-DSM. In addition, the height of the winter-DSM in high-mountain regions in the southwest and

also close to cloud-covered regions are slightly underestimated by 0.05 to 0.1 m. These underestimations caused a few negative snow depths on pixels covered by a thin snowpack in extremely steep areas. The classification of snow-covered pixels worked satisfyingly, whereby separate

pixels of sprinkled snow or snow mixed with soil of wet-snow avalanches are still falsely classified.

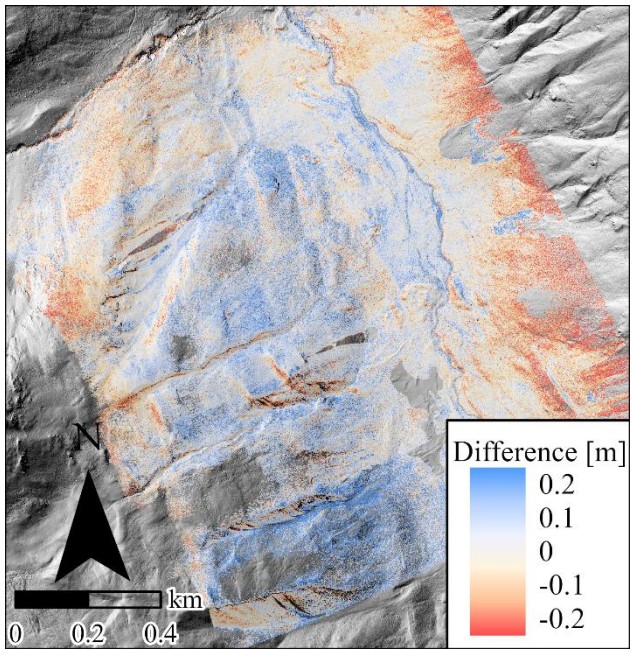

**Figure 8.** Difference calculation of the DSMs around the Schürliaalp derived from Ultracam data and UAS in 2018.

### 5.1.3 2019

The occurrence of technical errors on the airplane prevented the capture of the NIR-band, which can decrease the successful reconstruction of low-contrast snow surfaces and then, accordingly, also the accuracy of the DSM. However, except from

selected areas in north-faces, no significant influence on the reconstruction or accuracy of the DSM was determined. This statement is confirmed by evaluations of the RMSE of the check points (0.07 m) and the RMSE of the manually digitalized



snow-free areas (Filtered = 0.11 m) (Table 5; Fig. 10). Below the treeline, the height of the winter-DSM was slightly overestimated by 0.05 m. Using the snow-mask based on manually determined thresholds also led to a high-quality grade of more than 99 % correctly classified pixels (method described in section 4.2.1). The method used only had issues in snow

mixed with soil, on frozen rocks and in extremely tracked areas.

### 5.1.4    2020

The capture of the aerial images in 2020 was characterized by perfect acquisition conditions during the Ultracam flight. Consequently, the winter-DSM in 2020 evince a very high accuracy of around 0.1 to 0.15 m. This accuracy was determined by the use of a high number of well-distributed check points, which show a RMSE of 0.07 m (Table 5). The RMSE values of

method 4 indicate a similarly high accuracy (RAW = 0.19, Filtered = 0.13). Furthermore, the large snow-free areas in 2020 enable a representative accuracy assessment of method 5 which considers deviations in different elevations and slopes. The RMSE of the filtered deviations (0.18 m) in combination with the NMAD (0.16 m) shows the reliability of the winter-DSM in the entire study area and also in complex terrain. The deviations of M5 in extremely steep areas exceeding 40° (Filtered RMSE = 0.3 m) confirmed, that the inaccuracies of the winter-DSM increase with rising steepness.

### 5.1.5    2021

In 2021, the surface above around 1800 m a.s.l. was covered by a new snow layer, which caused less contrast during the Ultracam recording. Despite, these difficult conditions the check points indicate a similarly high accuracy like in 2020

(RMSE = 0.12 m). The RMSE (RAW = 0.14, Filtered = 0.12) values of the comparison between the DSMs derived from Ultracam and UAS also show satisfying results, with a slight tendency to underestimate the winter-DSM (Fig. 9). The underestimation is characterized by a negative median

(Filtered = -0.09). The median values of method 4 (RAW, Filtered = 0) and method 5 (RAW, Filtered = 0) indicate however, that this underestimation is a local problem and not valid for the entire study area. The low RMSE value calculated with method 4 (Filtered = 0.11 m) and 5 (Filtered

= 0.16 m) demonstrates the high accuracy of the snow depths (Table 5). Additionally, partly cloud-covered areas led

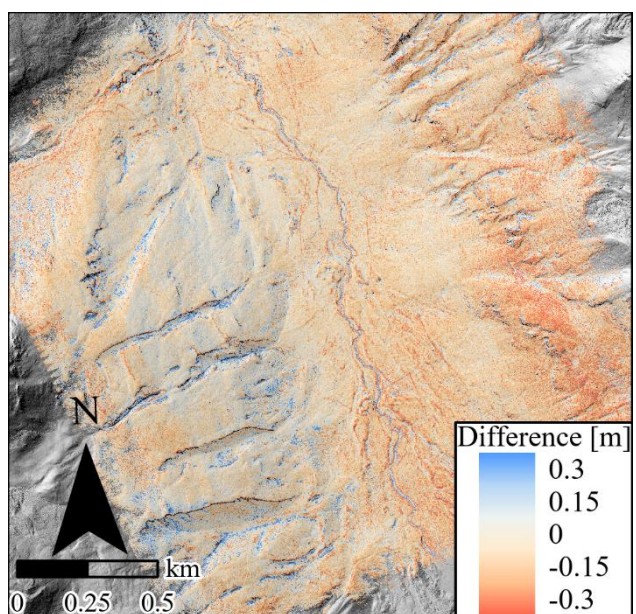

**Figure 9.** Difference calculation of the DSMs around the Schürliaalp derived from Ultracam data and UAS in 2021.

to no significant increase of the dispersion, which is shown in the low NMAD values of method 4 (RAW = 0.09) and method 5 (RAW = 0.15).





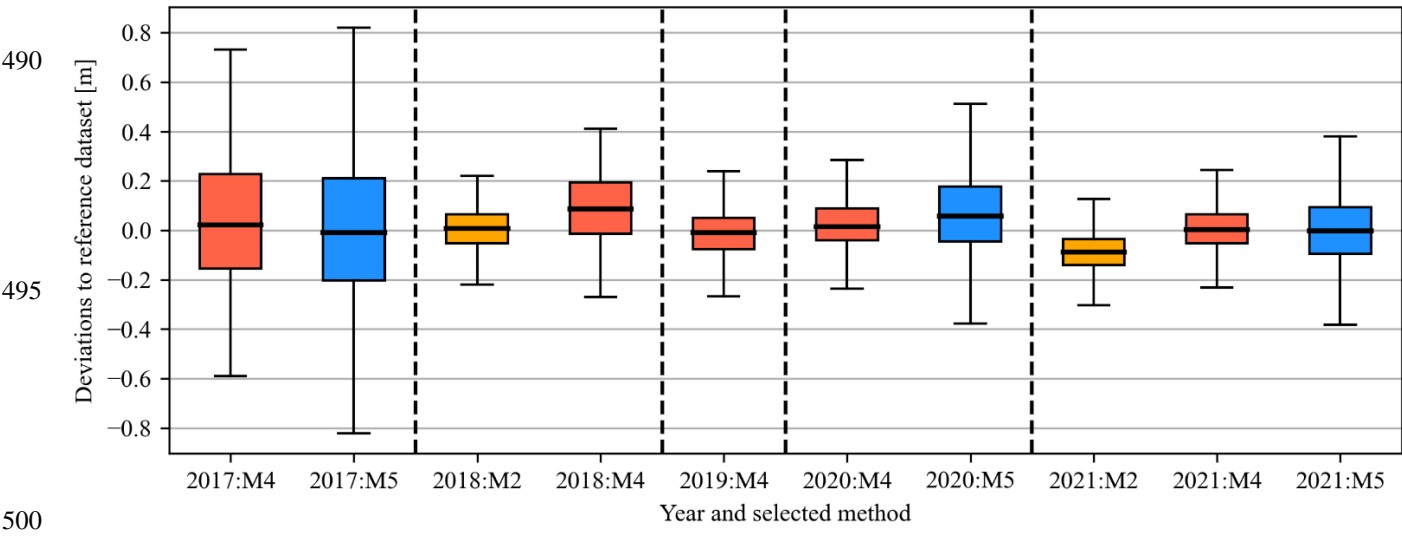

**Figure 10.** Box plots of the filtered deviations (MBE ± 2SD). Used Methods: M2 (orange), M4 (red) and M5 (blue) for each year.



**Table 5.** Overview and comparisons of the quality grades of the winter-DSMs; Column "Filtered" excluded outliers (MBE ± 2 SD).

| | | M1: Check Points | M2: UAS | | M4: Comparison ALS (streets) | | M5: Comparison ALS (snow-free pixels) | |
|---|---|---|---|---|---|---|---|---|
| | | RAW | RAW | Filtered | RAW | Filtered | RAW | Filtered |
| **2017** | MBE[m] | ✗ | ✗ | ✗ | 0.07 | 0.045 | -0.02 | 0 |
| | SD[m] | ✗ | ✗ | ✗ | 0.33 | 0.26 | 0.42 | 0.3 |
| | **RMSE[m]** | ✗ | ✗ | ✗ | **0.33** | **0.26** | **0.42** | **0.3** |
| | MdBE [m] | ✗ | ✗ | ✗ | 0.03 | 0.02 | -0.02 | 0 |
| | NMAD [m] | ✗ | ✗ | ✗ | 0.28 | 0.28 | 0.32 | 0.3 |
| | Number measurements | ✗ | ✗ | ✗ | 347'068 | 336'135 | 15'003'508 | 14'355'536 |
| **2018** | MBE[m] | ✗ | 0 | 0 | 0.07 | 0.08 | ✗ | ✗ |
| | SD[m] | ✗ | 0.12 | 0.09 | 0.16 | 0.14 | ✗ | ✗ |
| | **RMSE[m]** | **0.13** | **0.12** | **0.09** | **0.18** | **0.16** | ✗ | ✗ |
| | MdBE [m] | ✗ | 0 | 0 | 0.08 | 0.09 | ✗ | ✗ |
| | NMAD [m] | ✗ | 0.09 | 0.09 | 0.16 | 0.15 | ✗ | ✗ |
| | Number measurements | 10 | 7'471'650 | 6'949'690 | 132'558 | 127'191 | ✗ | ✗ |
| **2019** | MBE[m] | ✗ | ✗ | ✗ | ✗ | 0 | ✗ | ✗ |
| | SD[m] | ✗ | ✗ | ✗ | ✗ | 0.11 | ✗ | ✗ |
| | **RMSE[m]** | **0.07** | ✗ | ✗ | ✗ | **0.11** | ✗ | ✗ |
| | MdBE [m] | ✗ | ✗ | ✗ | ✗ | 0 | ✗ | ✗ |
| | NMAD [m] | ✗ | ✗ | ✗ | ✗ | 0.09 | ✗ | ✗ |
| | Number measurements | 6 | ✗ | ✗ | ✗ | 25'944 | ✗ | ✗ |
| **2020** | MBE[m] | ✗ | ✗ | ✗ | 0.06 | 0.04 | 0.09 | 0.07 |
| | SD[m] | ✗ | ✗ | ✗ | 0.18 | 0.12 | 0.27 | 0.17 |
| | **RMSE[m]** | **0.07** | ✗ | ✗ | **0.19** | **0.13** | **0.28** | **0.18** |
| | MdBE [m] | ✗ | ✗ | ✗ | 0.02 | 0.02 | 0.07 | 0.06 |
| | NMAD [m] | ✗ | ✗ | ✗ | 0.10 | 0.09 | 0.17 | 0.16 |
| | Number measurements | 40 | ✗ | ✗ | 221'087 | 214'114 | 30'933'482 | 29'522'927 |
| **2021** | MBE[m] | ✗ | -0.08 | -0.08 | 0.03 | 0.01 | 0.02 | 0 |
| | SD[m] | ✗ | 0.12 | 0.08 | 0.18 | 0.11 | 0.27 | 0.16 |
| | **RMSE[m]** | **0.12** | **0.14** | **0.12** | **0.19** | **0.11** | **0.26** | **0.16** |
| | MdBE [m] | ✗ | -0.09 | -0.09 | 0 | 0 | 0 | 0 |
| | NMAD [m] | ✗ | 0.08 | 0.08 | 0.09 | 0.09 | 0.15 | 0.14 |
| | Number measurements | 19 | 16'386'474 | 15'987'661 | 227'907 | 217'453 | 6'342'785 | 6'141'131 |





## 5.2    Snow depth maps

Despite different and complex acquisition conditions (section 3.1) as well as some extremely steep and complex areas, on average, more than 99 % of the snow depth values in the requested and not masked areas were reconstructed. Only in the winter-DSM from 2017, the image matching failed in few overexposed or shaded regions due to poorer radiometric characteristics of the sensor Vexcel Ultracam X. The high rate of success enabled the spatially continuous snow depth mapping of the region around Davos. The spatial resolution of the maps (0.5 m) and the orthophotos (0.25 m) provide an excellent

overview of the snow depth distribution within the study area. The existing level of detail shows numerous small-scale features over such a large area and demonstrates the high variability of snow depths even within small distances. Special characteristics of the study area are the strongly varying average snow depths and snow cover. The average snow depth values of the selected years ranged from 1.29 m in 2017 to 2.36 m in 2019 (Table 6). In particular the comparison of the snow depth maps from

2019 and 2020 is suitable to visualize significant differences of the snow depth distribution and related features (Fig. 11). When exemplarily comparing 2019 with 2020, we found that in 2019 the study area was almost completely snow-covered, exhibited numerous regions with high snow depths

**Table 6.** Overview average snow depths [m] and standard deviation [m] of each annual snow depth map.

| Year | 2017 | 2018 | 2019 | 2020 | 2021 | Mean |
|---|---|---|---|---|---|---|
| **Average** | 1.29 | 1.50 | 2.36 | 1.42 | 1.71 | 1.66 |
| **SD** | 0.87 | 0.83 | 1.33 | 1.01 | 1.11 | 1.03 |

exceeding 3 m and was characterized by the occurence of many slab avalanches. In contrast to that, the average snow depth in 2020 was considerably lower with a mean value of 1.42 m, the area was characterized by often snow-free slopes below 2400 m a.s.l. in southern aspects as well as numerous glide-snow and wet-snow avalanches. In general, for 2020, the aspect-dependence of the HS was more decisive than in 2019.

However, despite the high difference of the average snow depth values between these two years, similar patterns regarding

the relative snow depth distribution and occurrence of special features are identifiable. In general, the HS in both years increases with rising elevation until a certain level close to ridges or peaks. Higher snow depths are more frequently found on northern aspects compared to south-facing slopes which shows the aspect-dependence of snow depths. Furthermore, higher and lower relative snow depths of both snow depth maps occurred at similar locations (Fig. 11).





**Figure 11.** Comparison of an extent of snow depth maps from 2019 and 2020 during corresponding peak of winter. The black polygon shows the location of the selected small catchment for a more detailed comparison between the available snow depth maps (Fig. 12) (map source: Federal Office of Topography).

We further investigated the snow distribution patterns between the years by looking at relative snow depth distribution. The normalized snow depth values of each year were calculated by the relation of the HS in contrast to the average snow depth of the selected area in the corresponding year. In comparison to absolute snow depth maps, the normalized snow depth maps have the advantage of being independent from differences in the average snow depth between the years, which enables a better overview of the actual snow depth distribution. As depicted exemplarily in Fig. 12, we observed similar patterns for the distribution between all years. Generally, higher relative snow depths often occurred at deposition zones of avalanches, along terrain edges in wind-protected zones and within sinks, lower snow depths were frequently observed on slopes exceeding 35°, in wind exposed areas and in the release zones of avalanches (see also 6.2.2.). However, a few features such as selected avalanches only arose in separated years.





**Figure 12.** Comparision of the normalized snow depth maps of the 5-year period (2017 – 2021) during peak of winter in a small catachment (3 km²) close to Börterhorn. Numbers in the hillshade locate different special features, which demonstrate the existing grade of detail: 1. Filled small brooks; 2. Filled drain in extremely steep area close to Börterhorn; 3. Conspicous cornice between Tällifurgga and Witihüreli; 4. Cornice between Wuosthorn and the Börterhorn; 5. Deposition zone of avalanches.





Subsequently, we investigated the occurrence of further special features in the entire study area. The detailed detection of numerous avalanches by means of the snow depth maps and corresponding orthophotos is a salient characteristic of the data. In particular, glide-snow avalanches, striking slab-avalanches and deposition zones of wet-snow avalanches can be identified

which enables further research on the avalanche activity and characteristic of the corresponding period (Fig. 13).

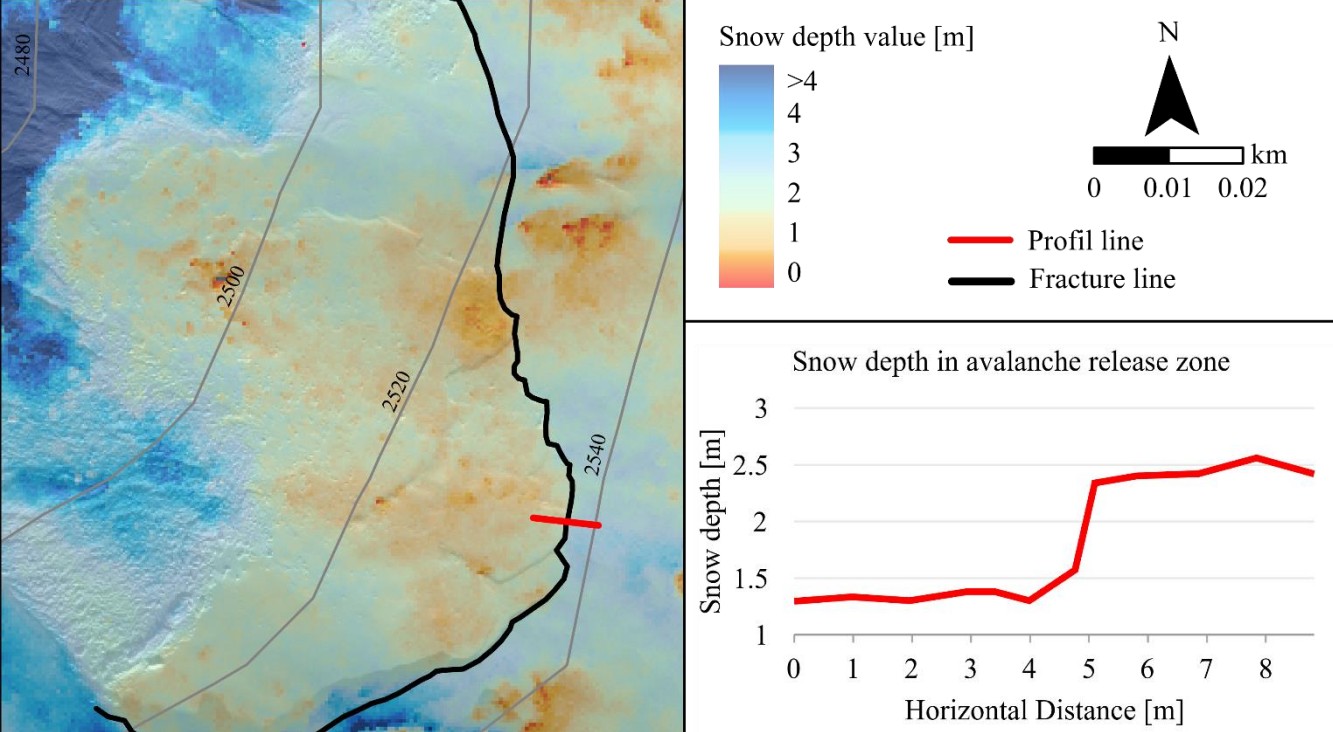

**Figure 13.** Overview of the snow depth distribution of a release zone of a slab-avalanche close to the Börterhorn (Dischma Valley) captured by the Ultracam in 2019; red line symbolizes a height profile which shows the course of snow depth values vertical to the fracture line; prominent difference of the snow depths indicates the release height of around 0.95 m.

**6    Discussion**

In this study we processed the data of annual Ultracam flights from 2017 to 2021 to snow depth maps. We investigated the necessary processing steps to derive accurate winter-DSMs, to create and apply required masks and the characteristics of the resulting snow depth maps. We analysed the accuracy and validity of the resulting winter-DSMs with the help of an accuracy assessment. In this section, we discuss the obtained results.

**6.1   Processing of snow depth maps**

This study focussed on the accurate and consistent processing of large-scale snow depth maps under different acquisition conditions with the new Vexcel Ultracam sensors over a period of five years. We have ascertained a significant quality increase from the Vexcel Ultracam X to the successor Eagle M3 due to its better GSD using the same flight altitude and better





radiometric characteristics. Data from the Ultracam X in 2017 exhibited errors in the NIR-band, which complicated the

classification of snow-free pixels. Additionally, the RGB-sensor of the Ultracam X was partly overexposed, hence Agisoft

Metashape had problems to find matching points in sunny snow areas. The accuracy assessment in 2017 also shows that the

RMSE of the winter-DSM, with around 0.25 m, is significantly poorer than in the following years (Table 5; Fig. 10), which

is no noticeably advance compared to other studies (Bühler et al., 2015; Nolan et al., 2015).

With the new Ultracam Vexcel Ultracam Eagle M3 sensor, Agisoft Metashape was able to reconstruct almost the complete

surface, even in heavily shaded areas, on surfaces covered by fresh snow as well as partly cloud-covered regions. Only in

completely cloud-covered areas, Agisoft Metashape had problems to reproduce the entire surface. This successful processing

is a significant progress for photogrammetric snow depth mapping compared to snow depth maps from previous studies

(Bühler et al., 2015; Nolan et al., 2015). Using the Ultracam Eagle M3 also resulted in a considerable better GSD under similar

flight altitudes compared to current studies (Meyer et al., 2021). The better GSD enables the exact processing of snow depth

maps with a spatial resolution of 0.5 m, which resulted in a better spatial resolution than previous large-scale snow depth

maps. The performance of extensive and significant accuracy assessments has shown a high reliability of the processed winter-

DSM based on the Vexcel Ultracam Eagle M3 sensor with an accuracy of approximately 0.1 to 0. 15 m (Fig. 10). The accuracy

assessment of the reference ALS-DTM compared to reference points (RMSE 0.03 m) has also demonstrated its high reliability.

Therefore, the accuracy achieved in the winter-DSMs corresponds approximately to the actual accuracy of the calculated

snow depth values. These accuracies of the snow depths match with the best results in Eberhard et al. (2021) and Meyer et al.

(2021) and fulfil the requirement of exact snow depth mapping.

Accordingly, despite the difficult and varying acquisition conditions, which often occur in high-mountain regions, the quality

grades of the snow depth maps demonstrate the robustness of the workflow used in this study. In addition, excellent acquisition

conditions such in the year 2020 resulted in no significant improvement of the quality metrics.

A crucial disadvantage of this workflow compared to the one of  Meyer et al. (2021) is the necessity of 2 to 5 GCPs. In the

presenting study area, the effort to measure the GCPs was low due to the vicinity to the WSL Institute for Snow and Avalanche

Research SLF (SLF) Davos. To capture remote regions, the exact processing of Ultracam data without GCPs would be

necessary. A promising approach could be the use of a global coordinate system, but first experiences have shown, that the

accuracy is considerably lower and less reliable compared to our workflow. Under consideration of this limitation, the

procedure is applicable on different study areas.

All computed metrics demonstrate an equal accuracy of our snow depth mapping compared to the one of an ALS (Deems et

al., 2013; Painter et al., 2016), which would be connected to considerably higher costs (Bühler et al., 2015). The crucial

disadvantages of this photogrammetric method are still the problems in high vegetated and especially heavily forested areas.

Currently, there are different approaches to capture the snow depth with photogrammetric methods within forested areas (i.e.

Broxton and van Leeuwen, 2020; Harder et al., 2020), but the dense forests in steep slopes around Davos impeded the accurate

and reliable recording of these areas yet. To counteract wrong values in those problematic areas, different masks were used

to increase the reliability of the snow depths.





This unique approach was previously applied in Bühler et al. (2015), but the algorithm used in our study is considerably improved and more reliable. Another important characteristic of these masks is the almost automatic and consistent
processing. Only the selection of streets and striking rivers requires manual interaction. In general, the masks are characterized by a high accuracy, but also exhibit few errors and limitations. In total, the percentage of incorrectly masked areas amounts less than 1 %, which is considered satisfactory for the accurate and automatic processing. The errors include snow-covered pixels in heavily shaded areas or snow mixed with soil falsely classified as no snow as well as not considered single-trees, newly-built buildings or environmental changes through mass-movements. As those changes are inevitable in the existing
structure of our study and the values account for a small proportion of our masks, we assess their effects as negligible. As for transferability to other study areas, the implementation of the masks depends on the availability of an exactly classified ALS data of the snow-free terrain. This is a crucial limitation of the method we used and would need adaption and evaluation of masks derived by other data sources.

## 6.2    Applications

The remarkable characteristics of the snow depth maps and the corresponding orthophotos enable new possibilities for various applications in science and practice. In particular, the assessment of natural hazards, research on snow depth distribution processes and snow-hydrological models as well as other measurement methods can benefit from this unique data and the method used. In the following we would like to discuss the relevance and potential impact of our work on selected applications.

### 6.2.1    Natural hazards

The investigation of natural hazards such as snow avalanches or snow loadings on buildings can benefit from the presented snow depth maps and the approach applied. Studies of Bühler et al. (2019), Hafner et al. (2021), Eckerstorfer et al. (2019) and Leinss et al. (2020) have already demonstrated the importance and the limitations of manual as well as automatic large-scale avalanche mapping with satellite data. On a smaller scale Korzeniowska et al. (2017) proved the automatic detection of avalanches on the basis of orthophotos derived from airborne photogrammetry (sensor ADS80). Due to better radiometric
characterisations and better spatial resolution of our orthophotos, even more details could be obtained than in previous studies. As exemplary shown in Fig. 13, the high-resolution orthophoto would allow for the exact identification of snow avalanches and associated release zones and deposition zones over larger regions. In numerous cases, the detection of the fracture line of an avalanche is possible as well. Consequently, the snow depth distribution around fracture line of the avalanche can provide meaningful information about the release height and therefore the release volume.
However, release zones covered by new or wind-drifted snow or avalanches released in extremely steep and complex terrain can lead to difficult identification or inaccuracies of the release height. Deviations of the snow height exceeding 0.3 m are unsuitable for the correct assessment of release zones. Nevertheless, these snow depth maps are the first data, which enable the determination of release heights of distinctive avalanches within larger regions. The examination of release heights with



remote sensing methods is only sparsely explored. Only individual studies with UAS were able to accurately identify the
release height so far (Bühler et al., 2017; Proksch et al., 2018; Souckova and Juras, 2020).

Furthermore, the assessment of snow volumes in release and deposition zones on the basis of snow depth maps and
orthophotos facilitate the research on avalanches. Studies with UAS have already demonstrated the high importance of these
measurements (Bühler et al., 2017; Eckerstorfer et al., 2016). The crucial advantage of our procedure compared to previously
performed studies with the UAS is the cover of larger areas during periods with high avalanche activity. However, the
necessity of the flight permission and the weather-dependence often prevents short-term missions.

The assessment of other hazards such as snow loading on buildings or flooding caused by rapid snow melting can also be
assisted by large-scale snow depth maps. For the determination of snow loading on buildings, an adaption of our workflow
would be required by using the DSM of the ALS as a reference dataset and not masking settled areas. Therefore, our accurate
snow depth mapping facilitates the assessment of dangerous snow loading on roofs.

**6.2.2    Snow depth distribution**

Our snow depth maps could be a key for the better understanding of snow depth distribution in alpine terrain, which influences
numerous sectors. The results presented in Fig. 11 and Table 6 show the strongly varying average snow depths, which
corresponds to the observations of the snow depth maps by Marty et al. (2019) and point out the added value of annual snow
depth maps. Despite the high difference of the average snow depths, we identified a similar relative snow depth distribution,
which was exemplary shown in the normalized snow depth maps in Fig. 12. Consequently, the relative snow depth distribution
between different years is almost independent of the average snow depths with the exception of separate avalanche depositions
zones and selected special features as they do not occur every year.

Since different publications had already described different snow depth distribution patterns, they will be discussed on the
basis of our results. The studies of Grünewald et al. (2014) and Prokop (2008) found, that snow at wind-exposed and steep
areas is relocated to flatter areas and sinks. Our results confirm these observations. For example, small creeks in high-mountain
catchments can be identified on the snow depth distribution in our snow depth maps, because the creeks are filled up with
snow and exactly these pixels exhibit higher snow depths than surrounding areas. Similar features can be recognised in
drainage channels in extremely steep slopes.

Bühler et al. (2015) and Schirmer et al. (2011) recognized the permanent occurrence of cornices at the same ridges around the
Wannengrat within our study area in two different years. Our data can verify the formation of this cornice in further years. In
addition, we determine the persistent formation of cornices in all years of the time-series at numerous ridges. These cornices
lead to considerably higher snow depths on the same side in each winter. Furthermore, different studies (Peitzsch et al., 2015)
have already suggested that the location of glide-snow avalanches is often similar between winters. The location of the
avalanches in our snow depth maps between the years show, that they are characterized through a high conformity.
Correspondingly, these observations can reinforce previously findings of the location of avalanches.



These observations concerning the relative snow distribution correspond to the results of Schirmer et al. (2011) and Wirz et al. (2011), which already found higher and lower relative snow depths on the same locations within a winter, respectively. However, all these studies were either limited to only one year (Schirmer et al., 2011; Wirz et al., 2011) or the accuracy and the spatial resolution of the snow depth maps (Bühler et al., 2015) complicated the investigation of snow depth distribution

patterns. Therefore, our snow depths are the first time-series which enables the extensive comparison of snow depth distribution between different years on a large-scale. These new possibilities lead to the confirmation of different theoretical approaches, which assumed the snow depth distribution is more dependent on terrain characteristics than on the weather conditions of a certain year. This revelation opens new possibilities for the modelling of snow depths over large regions.

### 6.2.3    Validation dataset

Different studies have already benefited from the existing unique time-series of large snow depth maps (ADS sensor) processed by Marty et al. (2019), starting in 2010. However, the inaccuracies and the lower reliability of these snow depth maps also limited the validation and evaluation of other studies. In particular, deep learning approaches or studies which are calibrated by exact reference data can now benefit from our improved quality. Therefore, it is to be expected, that the data will be applied in numerous studies. For example, the snow depth maps serve as training dataset for a running project to

improve the modelling of the daily snow depth distribution in Switzerland. Without our data, the model was not able to represent the snow depth distribution in complex terrain. In addition, our data could validate or evaluate numerous projects in conjunction with hydrological and snow models (Helbig et al., 2021; Richter et al., 2021; Vögeli et al., 2016), wind-drift models (Gerber et al., 2017; Mott et al., 2010; Schön et al., 2015), automatic detection of avalanche release zones (Bühler et al., 2018; Bühler et al., 2022) and further snow depth models or snow depth measurements on the basis of satellite data

(Leiterer et al., 2020; Wulf et al., 2020).

### 7    Conclusions

In this study we present the development, validation and application of a consistent and robust workflow to process aerial imagery from the state-of-the-art survey camera Vexcel Ultracam to produce reliable snow depth maps. We demonstrate its capability to capture large areas covering more than 100 km² under optimal as well as suboptimal acquisition conditions

(varying illumination, clouds, new snow cover, the absence of the NIR-band). The accuracies of our snow depth maps (RMSE: 0.15 m, Ultracam Eagle M3) are similar to results achieved with ALS and fulfil the requirements for meaningful, spatially continuous snow depth mapping in complex terrain. The metrics are calculated applying an extensive accuracy assessment with check points, comparisons to UAS-derived DSMs and the evaluations of snow-free pixels, revealing a very high quality even within steep terrain. The reliability of our maps allows for the comparison of the snow depth values within a 5-year

period, which have shown that despite large differences of the average snow depth, the relative snow depth distribution and the formation of small-scale features is similar throughout the years.

Restrictions of the data and its acquisition are the relatively high data acquisition costs (approximately US$ 20'000 for 300 km$^2$) and the availability of a piloted aircraft and corresponding permissions. In addition, the developed procedure is limited

by widespread low clouds, areas with high vegetation such as forests and bushes, the availability of accurate snow-free DTMs and powerful hardware with large storage for processing. Even though accurate GNSS and IMU data is available from the airplane, one to five ground control points (GCPs, distribution is not important) as well as the almost automatically calculated masks are essential to achieve reliable results.

In particular, the high spatial resolution of our maps (0.5 m) and orthophotos (0.25 m) in connection with the achieved accuracy offer the possibility to better understand the complexity of snow depth distribution in high-mountain regions. Based on the presented products, models of water stored in the snowpack (SWE) can be evaluated and improved, this is crucial for hydropower generation. New approaches to map snow depth with optical and radar satellites from space can be evaluated. Also, the investigation of snow avalanches benefits from such data. Several running research projects are already applying

our maps for validation. We expect that our data will become a key for numerous new findings in snow science in the coming years.

The improvement of photogrammetry within areas covered by alpine forests would be a significant step forward to equalize the advantages of ALS. Our data has already been used to validate and evaluate different research projects. Our data allows

for the extrapolation of the snow depth distribution from small areas, mapped for example by UAS, to the scale of large catchments. To further enhance the value of photogrammetric snow depth mapping, the current time series will be extended into the future. Together with datasets acquired from 2010 to 2016 with the ADS sensor within the same region, we established an unique eleven-year snow depth time series.


*Data availability*. The datasets used in this study will be published in ENVIDAT (https://www.envidat.ch) with the final publication of this study.

*Author contributions*. YB and LB designed the study. YB, AS, EH and LAE performed the fieldwork. LB processed the data
with inputs from MM and YB. LB and YB prepared the manuscript with contributions from all co-authors.

*Competing interests*. The authors declare that they have no conflict of interest.

*Acknowledgements*. We would like to thank the Swiss National Science Foundation (SNF; Grant N° 200021_172800) for
partly funding this project. We also thank the assistants for their help during the various fieldworks.

*Financial support*. This research has been partially supported by the Swiss National Science Foundation (SNF; Grant N° 200021_172800).




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
