# Peer review of "Spatially continuous snow depth mapping by airplane photogrammetry for annual peak of winter from 2017 to 2021 in open areas"

_The Cryosphere, 2022_

## Author Comment (AC1)

Dear anonymous reviewer. Thank you for your rigorous and detailed review.

While a lot of your comments are helpful to improve our manuscript we disagree with your main criticism about our work:

1.) You state that validation surface elevations are not the same as validating snow depths. While the statement itself is true, our own and other groups (Deems and Painter, 2006; Vander Jagt et al., 2015; Bühler et al., 2016; Harder et al., 2016; De Michele et al., 2016; Goetz and Brenning, 2019; Eberhard et al., 2021) proved that precise surface models are a very good proxy for snow depth values. A lot of time and effort was taken by various groups to collect manual snow depth measurements and to support this conclusion. Furthermore, the meaningfulness of a comparison of spatial coherent snow depth maps with isolated point measurements (acquired manually with probes or at weather stations) is very limited. Therefore, we are confident that the validation as presented in our paper is the most meaningful way possible to validate the results.

2.) We do not agree with the statement that photogrammetric snow depth mapping over large areas is an established method. In particular, snow depth mapping based on piloted aircrafts has only been investigated by a few groups including ours (Smith et al., 1967; Cline, 1994; Bühler et al., 2015; Nolan et al., 2015; Meyer and Skiles, 2019). None of these studies reached a sub-meter spatial resolution and more importantly, produced a time series of snow depth maps under different acquisition conditions. We are therefore convinced that applying a state-of-the-art surveying camera and evaluating the results of the time series is a very relevant contribution for the scientific community.

3.) You state the value of our products are very limited because some areas (mainly forested and settled regions) are excluded. We are well aware of the limitations of photogrammetry compared to LiDAR. However, for alpine regions, where only small portions of the landscape are covered by the excluded surface types, the value of our dataset is very high. A confirmation for this is the high request for our dataset for different research and application projects where the data is already applied. To our knowledge the datasets presented in this study are the only ones covering a large area (> 100 km²) with high spatial resolution (submeter) over different years, except the LiDAR based Airborne Snow Observatory (ASO) data in the US (Painter et al., 2016). Compared to ASO our approach has better spatial resolution and is much more economic.

4.) You state that we only got good results because we masked out all snow depth values with a poorer accuracy. We disagree with this comment. We are not cherry-picking good snow depth values. There is a crucial difference between filtering out poorly calculated values and being aware of the areas in which photogrammetric methods often struggle and masking them out in a structured way.

5.) You state that data acquisitions with crewed aircrafts are very expensive, and it does not matter if it is with LiDAR or photogrammetry. Based on a recent quote (2018) and discussions with data providers, digital photogrammetry is at least 40 % cheaper than covering the same area with airborne LiDAR. For us this is a very significant price difference, even without processing costs, which would also be higher for LiDAR.

However, we will consider following things of the major points for our revised version:

We will change the title of the paper to: "Spatially coherent snow depth mapping by airplane photogrammetry for annual peak of winter from 2017 to 2021 in open areas". We do not see a problem using LIDAR as the snow-free model instead a summer-DSM derived from photogrammetry, because the focus was the processing of the winter-DSMs under different acquisition conditions.
However, we will add the limitation of our method used for the transmission on another study area, where no accurate LIDAR is available.
In addition, we will replace the automatic workflow by a consistent workflow. We will emphasise the application of our data for snow-hydrological modelling more, for example to correct modelled snow depth values.

In the following we will answer the specific comments (the numbers correspond to the lines in our manuscript). The intended arguments are your comments and our response follows:

> *(13/14) - Echoing comment above, broadly snow depth is valuable for natural hazards, but I'm not sure this is suitable motivation for the results presented in this paper.*

We disagree with this comment. Our snow depth maps enable the investigation of snow depth distribution pattern in general. Information on snow depth distribution, especially in high-mountain areas, where continuous snow depth mapping is possible, is crucial for the assessment and the understanding of various natural hazards.

> *(18) It's actually not considerably more expensive in terms of day to day operations, especially now that there are private companies like Airborne Snow Observatories, Inc streamlining data processing and collection. And again, not sure that's a reasonable argument in this case, when photogrammetry flights are so expensive only one per season can be afforded.*

As already said in the main points, airborne LIDAR is in Switzerland at least 30% more expensive than covering the same area with aerial photogrammetry. Our method used is still very expensive, but the best procedure in terms of the cost/accuracy ratio.

> *(24) It needs to be clarified here and elsewhere that this is open snow mapping - you can't call it a snow depth map when you explicitly exclude areas where there is snow.*

We will modify snow depth mapping to snow depth mapping of open areas.

> *Introduction*

*37-43 Not relevant background for the presented work - the temporal scale of one flight per season at peak is not all that useful for dry slab avalanches, that typically occur before peak snow depth, during the accumulation season, and you can't tell differences in distribution due to environmental processes with only one flight per season. This study also masks out urban areas, so I'm not sure why snow loads on buildings are relevant.*

We disagree with most of your comment. Of course, having only one flight per season has some restrictions, but as every winter is different our five year time-series does capture a variety of conditions which allows us to draw overall conclusions. We have, for example in 2019, captured numerous dry slab avalanches after a period with high avalanche danger (danger level 4 of 5).
The time-series of the normalized snow depth maps enables research on differences due to environmental processes between the years. For example, we have shown that the accumulation locations of wind-drifted snow or the aspect of cornices show a consistent pattern between the years. While your statement about snow loads is itself right, a small adaption of our general workflow is sufficient to capture snow loads. Using the digital surface model (DSM) of the LIDAR as reference data instead of the digital terrain model (DTM) and not masking out the urban areas enables us to calculate the snow depth values on the roofs.

*45-46 Again, relevance, this paper does not map catchment SWE. It maps snow depth in the open part of the catchment.*

You are right, that the paper does not map catchment SWE. But, it provides almost spatially continuous snow depth maps of a high-mountain catchment as well as open areas. Therefore, our data can serve as a bases for the calculation of the SWE through a multiplication of the measured snow depth values with a modelled snow density. Combining our snow depth with a modeling approach for SWE would be beyond the scope of this work as our main focus is to present a reliable method for snow-depth mapping over a large catchment. But this would be interesting to investigate in the future.

*66 If you exclude vegetated areas, urban areas, etc you are also not mapping continuous snow depth distributions*

Our maps provide an almost spatial continuous coverage in high-mountain areas and an approximately 66% coverage of the entire region (see Table. 3), which is a considerably higher coverage than using a GPR.

*68-69 Mentioning the need for a helicopter seems very specific to the cited study and out of place*

We will delete this sentence in the revised version.

*73 is ranging -> ranges*

We will change it.

*80 Is importance the right word here? Or has it just seen wider adoption because the data collection and processing is simplified in comparison to traditional photogrammetry.*

We will replace the word importance by:" the use of". The mentioned arguments such as the SIFT-algorithm, development of easy-to-use software and the use of drones should clarify the reasons for the increasing application of photogrammetric methods.

*83 Is it true that the accuracy mainly depends on the sensor and GSD? What about GCPs (refer to your own manuscript, also Goetz et al)? What about collection conditions?*

We have already shown that the accuracy mainly depends on the sensor and the GSD (Eberhard et al., 2021). Most of modern unpiloted aerial systems (UAS) have RTK and do not require additional ground control points (GCP) to georeference the model. For example, the processing of our UAS data in 2018 and 2021 was carried out without the implementation of GCPs. However, the check points we have used have shown very good accuracy (for details see line 187).
Poor collection conditions can still be a limitation and influence the accuracy, but all measurements are influenced by very poor collection conditions. In addition, the following sentence in the manuscript clarifies the limitations.

*91 This statement needs elaboration as it makes it sounds like NIR is required for large scale snow depth mapping.*

You are right, the availability of the NIR-band is not an essential requirement. However, the NIR-band can improve the accuracy of snow depth values (derived by optical satellite images) in low-contrast areas. We are therefore convinced that the NIR-band has an added value for snow depth mapping with satellite imagery. In our case 2019 without NIR has similar accuracy compared to the years with NIR band. We will adapt such and make it clear in the revised version.

*97-98 For the McGrath paper the RMSE was for the 8 m output products and the 0.3m is the GSD*

Thanks for the hint. We will replace spatial resolution by ground sampling distance (GSD). We will add that the data derived from WorldView-3 satellite was rasterized to an 8 m grid.

*99 what is meant by 'stimulates the application'*

We will replace stimulate by promote to clarify the statement.

*100-101 This should say the actual spatial resolution and the RMSE (or what was used for error measurement)*

*We can only give an overview in the introduction, detailed metrics for every method mentioned would be beyond the scope of this paper.*

*104-107 Think these numbers have changed quite a bit in the last seven years with increased used of airborne lidar, also, these both are still considered 'expensive measurements'*

See major points.

> *100-115 Overall this discussion is confusing and inconsistent. It sounds like cherry picking results from different papers - i.e. Nolan did get excellent accuracy but over a very small area, and Meyer could have produced the maps at higher resolution with the point density but the maps covered a very large area - so it would be helpful to be consistent with the literature description. For example, what cameras were used in other studies, what was the area covered and resolution, what was the uncertainty. Otherwise it's very challenging to understand how these studies compare to each other and to this one.*

We have chosen to give more details for the work that we found most relevant for our study, an introduction is usually not a comprehensive summary of detailed content of all cited work. Therefore, for the demanded details the reader must turn to the original work as this would be beyond the scope of our work. However, we will rephrase this section and add the type of camera, the covered area, the spatial resolution and achieved accuracy to each paper to enable a consistent comparison between the publications.

> *120 Can seven years, not even a decade, be considered long term monitoring?*

Combining the dataset from 2010 to 2016 with our dataset from 2017 to 2021 (2022) results in a 12-year time series, which to our knowledge corresponds to the longest available time-series about large-scale snow depth mapping. We consider 12 years long term from a remote sensing perspective in particular because we will extend this time series into the future.

> *Data and Sensor*
> *150 Can you provide more details? Is from a camera mounted GPS or from the airplane?*

To compress this section, we refer to the cited publication from Eberhard et al. (2021), which provide more details. However, we will add the information, that the GPS data came from the camera.

> *154 Did they actually end up being at peak? Although obviously easy to identify after the fact, it can be very challenging to predict the timing of peak. I assume this area has many automated snow depth sensors, so it should be easy to discuss how close to peak flights actually occurred.*

You are right, it is difficult to determine the peak of winter. It is also not trivial to discuss the actual peak of winter, because the peak of winter varies within such a large study area. We define the peak of winter as the date with the highest average snow depth value at an elevation band between 2000 and 2500 m. In addition, acquisition time also depends on weather conditions and flight permissions, so it is only the approximate date for the peak of winter. However, we checked the snow depth values at the Weissfluhjoch test site (2500 m) and according to this information, the flights performed were always within days of the actual peak of winter. We will add our definition of peak of winter to the revised version of our manuscript.

> *168 Are you inferring snow free photogrammetric elevations are not accurate?*

No. In our case, Swisstopo already carried out an airborne laser scanner (ALS) and provided the high-resolution and well-validated point cloud free of charge. If no ALS point cloud had been available, the recording of snow-free reference dataset derived from photogrammetry would have been also an option. As it was not necessary, we did not conduct an additional flight with a mounted camera during summer to reduce the costs.

> *172 More details are needed here. For example, were the 20 pts per square meter all ground returns? Also its bold to create a 0.5 DTM from a point cloud that has 0.2m horizontal accuracy, especially since only snow surface elevations are assessed, the accuracy assessment ignores errors in snow depth that comes from misalignment of the snow free and snow on surfaces.*

We will add that the point density of 20 points/m² corresponds to the average number of all points. We cited the accuracy given by Swisstopo, for details one would have to turn there. However, numerous applications of this ALS point cloud have shown, that Swisstopo underestimates the accuracy and accordingly, the actual accuracy is often considerably better. To investigate this, we compared 24 manual measured points with the derived height from the ALS (see chapter 5.1; Root mean square error RMSE: 0.03 m). We checked the snow depth maps on striking misalignments, which would lead to large inaccuracies in extremely steep areas. Due to satisfying results in these areas and the limited availability of stable as well as snow-free areas over the entire region in a few years, we decided to skip a further alignment between the snow-on and snow-free models, even though we consider that a valid processing step.

> *182 Can you quantify 'slightly positive temperatures' and what you mean by mentioning this, so it is not so vague?*

We will add the maximum zero degree level (2500 m) to clarify slightly positive temperatures. We will also add that we expected therefore only a low snowmelt as well as a low settling of the snowpack.

> *187 This is a result, how the numbers were arrived at should be discussed in the results and not presented as a fact in the methods*

We will move the RMSE values of the check points (CPs) to chapter 5.1 (Accuracy Assessment).

> *191 - 192 Can you explicitly mention that these reference points, located in easy to access terrain, are not representative of more complex terrain and therefore would impact the overall accuracy of the maps (which cannot be known without assessment points in that terrain)*

We will add that the check points were mostly located in flat terrain. It is well-known that point measurements cannot be compared to height values derived from raster in complex terrain.

> *193-194 Can these also be shown on the map?*

We thought about showing them on a map. However, visualizing the 40 points in an existing map (for example Figure 1) would have led to a loss of overview of the snow depth map extents. Furthermore,

due to the uniform distribution of the reference points over the entire area, we do not see a striking added value of showing them in an added large map. For the interested reader we will add an overview map to the Appendix in the revised version.

*209 What were the other coordinate systems?*

The other coordinate systems were WGS84 and the old Swiss LV03 horizontal coordinate system. However, we do not see an added value to mention them explicitly.

*Methods*
*I'm not sure what part of this is considered automated as is mentioned in the abstract*

We will replace automatic by almost consistent method to correct this.

*217 You can set the level of filtering? You used aggressive in 1.6, why not just lower the level of filtering in 1.7? This part opens up more questions than answers and could be excluded.*

Yes, you can set the level of depth filtering. We carried out numerous tries with different settings and Agisoft MetaShape versions. Based on our extensive experiences with snow surfaces, we mentioned that we processed the data with version 1.6 in combination with aggressive depth filtering to get the best results. Using version 1.7 and milder depth filter produced poorer results.
We will rephrase and shorten this section.

*4.1 Processing workflow - it is unclear on how the RMSE is arrived at. Since it seems like a confident result about only needing up to 5 points for 240 km^2, I think it would be worth spending the time to explain this accuracy assessment.*

We will add that the RMSE was calculated by the check points (CP), which were not used as GCP. However, to clarify this, we will cite the Agisoft manual, where the calculation of the RMSE based on CPs is well-described.

*270 What was the point density of the SfM DSMs? Since it was presented for lidar, seems fair to present it here. Also 'is often' -> often being*

We will add a subordinate clause including the point density of the winter-DSMs. They are ranging from 5 points/m² (2017) over 20 points/m² (2019 - 2021) up to 90 points/m² (2018). We will correct this grammatical mistake in the revised version.

*287-289 Good and bad results are important, just masking out all the bad results does not make for good science. Can you go into more depth about the snow depth overestimation?*

We will add the information, that the overestimation was mostly caused by the buildings and the forested areas. This is a crucial limitation of photogrammetry which we are very aware of and have mentioned several times. We have not chosen to get rid of bad results that way, but rather we masked

those areas out because it methodologically, and therefore scientifically makes most sense. For example, when subtracting the DTM from the winter DSM the values at the location of a building will be the building height plus snow on top. For details, also why we used a DTM and not a DSM as reference, see line 271.

> *Figure 4. If I read the methods correctly, you are taking the geo-referencing results out of Agisoft at face value and not assessing any co-registration differences between snow on and snow free. I think this is pretty important given the high spatial resolution to at least have an idea on how much they are off set - it is critical for accurate snow depth mapping. Also, you are mixing masks from two different data sources at very high resolution, and offsets would also be relevant during this process.*

You understood it correctly. As already said, the misalignment was checked in extremely steep areas. In addition, co-registration needs snow-free and stable areas, which were hard to find in some years, especially beyond settlement areas, extremely steep areas, or streets.
However, we will add this to the discussion about the limitations of our data and method in the revised version.

> *312 Highlighting that any workflow that contains a 'determined by an expert' step cannot be claimed to be automated*

We will adapt the declaration about the almost consistent workflow.

> *4.2.2 Outlier Mask - Why does the UAS map use a value of 3x the confidence interval and here it is set to 10m? Consistency would be valuable. Also, why do you need this mask if the snow depths and error statistics have a 'filtered' and unfiltered category. Think this makes the unfiltered category look better than it is.*

These are completely different maps and therefore different approaches. The UAS map represents the differences of the DSMs derived from UAS and the Ultracam flight. Correspondingly, no masks were applied because the data was generated with the same procedure, is therefore comparable and the summer-DTM causing the high deviations in forested areas and settlements was not involved. However, outliers occurred due to the delay of the airplane in 2018, other environmental processes and different capture angles of the corresponding sensors. Since high outliers have a high influence on the RMSE, only the recommended 95 % interval of all values (Hoehle and Hoehle, 2009) was used, which corresponds to mean +/- 2 standard deviations (filtered).
We did not filter the snow depth values depending on a statistical distribution. We defined areas, in which photogrammetric methods often have problems and masked them out. The outlier mask especially including not considered buildings or forested areas by the corresponding mask. The negative values are physically not possible, accordingly we had to mask them out. The filter based on the statistical distribution was only applied for the automatic accuracy assessment to represent the actual accuracy.

> *Table 3. A percentage map for all years would help to interpret the snow depth results and errors below*

Due to the large study area and the high spatial resolution, the calculation of these numbers requires days. Moreover, only the relative part of the outliers of each snow depth map would be an added value because the other masks are consistent and only vary due to different extents of the snow depth maps. The number of outliers increases in years with shallow snow depths, because negative snow depths occurred more often. Since 2018, 2019 and 2021 were snowier than 2020, the number of the calculated outliers in 2020 should be similar compared to the other years. Only in 2017, when the average snowpack was lower and the inaccuracies of the method used higher, probably more outliers occurred.

*366-367 This sentence reads like a contradiction. Extensive accuracy assessment, but no ground reference dataset?*

We suppose you meant 356/57 and not 366/67. We did perform an extensive accuracy assessment. Due to the lack of one spatially continuous reference, we instead used a combination of several analyses. The combination of those investigations yields an extensive accuracy assessment.

*370 Why are satellite paper citations backing up this argument?*

We cite Deschamps-Berger et al. (2020) and Marti et al. (2016) as they also used UAV-derived DSMs as reference to validate the processed snow depth values derived from satellite data. Through that we wanted to emphasize that it is widely used and has been done before.
For the methodology it makes no difference, if the snow depths were derived from airplane photogrammetry or satellite photogrammetry.

*386 This seems like a huge number of a static surface.*

Yes, large areas of the streets were covered by puddles caused by the snow melt on the day of acquisition but also from rain the day before (snowfall line at times over 2000m).

*389 - 390 Is this reasonable? The ALS has a vertical uncertainty of 0.1 m?*

Yes, since, no ground truth was available, we still wanted to quantify our results. We consider the proper description of the uncertainties of the reference datasets enough, especially as any dataset has their advantages and limitations. Therefore, an ALS with a vertical uncertainty of 0.1 m and better, can be considered as ground reference on snow-free and not high-vegetated areas.

*Results and Validation*
*5.1 Accuracy Assessment - Assessing the vertical accuracy is important, but what about the horizontal accuracy?*

You are right, the horizontal accuracy is also important. However, the horizonal accuracy strongly influences the vertical accuracy, especially in steep areas. Correspondingly, the given vertical accuracy contains the horizontal inaccuracies. In addition, it is difficult to exactly determine the horizontal accuracy. However, we will add, that the check points show a horizontal accuracy of approximately 0.1 m in the revised version. Whereby this calculated accuracy is also influenced by uncertainties of the GNSS and marks of the CPs.

*426 What is the definition of complex terrain? Vegetation? Steep areas? Could this also be affected by shallow snow depths?*

We will replace complex terrain by extremely steep and heterogenous terrain to clarify this in the revised version.

*Figure 8. Can you show location of GCPs? This looks like it could be systematic error that can be caused by the lack of well distributed points.*

As described in the cited publication from Eberhard et al. (2021), no GCPs were used for the correct alignment.

*456 Are these the M1 check points? Those were stated as not well distributed. How can this give confidence?*

Yes, they were only located in Davos and in the Dischma valley. Because they alone cannot provide a high confidence over such a large study area, we additionally carried out the manual selection of 25 000 pixels (M4) to ensure a meaningful accuracy assessment.

*457 Again, here, anything manually determined is an argument against the claim of 'almost automated'*

See major points.

*468-469 Not the first study to show this, citations could be useful here*

We will add the publications of Bühler et al. (2015) and Meyer et al. (2021), which already found out a lower reliability of snow depth values derived from photogrammetric methods in steeper areas in the revised version.

*527 Overall I find the presentation of snow depth results too general and shallow - the results are obvious and can be found in every other paper looking at snow depth distribution, for example, more snow on north facing slopes*
*Figure 12. Is there something off with the color scale? Is the deepest red color zero?*

With the time series and covering a large catchment we find it important to confirm previous studies and restate the obvious. We could go more into depth though, you are right. Consequently, we will describe in detail the findings on the snow depth distribution in the revised version.
For example, we will add the actual snow depth values of Figure 12 to enable a better overview of the differences between actual and relative snow depth distribution. In addition, we will add a further application of our snow depth distribution regarding the agreement between the measured snow depths at different weather stations to our average snow depth values derived from the snow depth maps. We will also add the application to find suitable locations for snow depth station which represent the snow depth distribution well in this region. More detailed and statistical investigations on the snow depth distribution and the influence of different terrain parameters will be performed in a subsequent study. This paper serves a bases for different projects and should explain the workflow used and the accuracy of the data.

*570 Meyer and Skiles, 2019 established that fresh snow surface elevations can be accurately mapped with photogrammetry*

Thanks for the hint. There have been several studies that have shown this applicability at small locations and in isolated points in time. However, the application to a whole catchment (250 km² compared to 3 km²) is new. Additionally, the flight altitude (4000 m above ground compared to 1500 m above ground), sensor used, and study setups are very different. We are able to confirm the findings of Meyer and Skiles (2019) and others for large-scale snow depth mapping.

*571 This isn't a problem with Agisoft - if you can't see the surface you can't reconstruct it with photogrammetry*

We will rephrase this sentence to clarify that is not a problem of Agisoft.

*577 'approximately 0.1 to 0. 15 m (Fig. 10)' these are filtered values*

That is right. However, we have extensively explained why the unfiltered values of M4 and M5 underestimate the accuracy. In contrary, the filtered values of M4 and M5 represent the actual accuracy of the winter-DSMs. The considerably better metrics of the manual digitalized areas in 2019 (M4) as well as visual checks also proved that in particular falsely selected pixels by our automatic accuracy assessment approach caused the poorer metrics of M4 and M5. Using only the check points, the manual digitalized areas or the UAV-derived DSMs, the metrics calculated do not require a further filter. These unfiltered values exhibit the accuracy of 0.1 to 0.15 m as well.

*580 - 581 This paper used the same data as Eberhard 2021? Seems like a circular reference, but also on Line 574, it said to have way better GSD. Why not better results?*

Yes, we used the same images as Eberhard et al. (2021), but our workflow was adapted using considerably less GCPs and an almost consistent workflow.
In our study we achieved an excellent accuracy of around 0.15 m. Correspondingly, there is not a lot of room for improvements, and it is also difficult to prove a better accuracy due to the limitations in the ground truth datasets. In addition, the accuracies depend on numerous and complex factors, which are different in Meyer et al. (2021). However, our better GSD led to a higher spatial resolution of the snow depth maps, which enables new and better investigations of small-scale features.

*582 -583 This is workflow is ...very standard? I'm not sure what this paragraph is trying to say. Everyone follows the same basic steps to get snow depth, and anyone can filter out poorly reconstructed areas to get good snow depths in open areas.*

Most steps of the workflow presented in Agisoft Metashape are well-known, but details often vary depending on the sensor used. Accordingly, to develop a consistent workflow which works under different acquisition conditions, for various years is a crucial step forward. To our knowledge there is no comparable dataset that exhibits such a high reliability which can directly be used without further post-processing for meaningful analysis or as input for various models. We are not cherry- picking good values- there is a crucial difference between filtering out poorly calculated values and being aware of the areas in which photogrammetric methods often struggle and masking them out in a structured way.

*588 -590 So we always need roads? This paragraph does not say much.*

No. However, to apply our method used in completely remote regions, further research on the necessity of GCPs would be required. In regions with large snow-free areas the co-registration approach could be helpful. In almost completely snow-covered areas, GCPs and accordingly, an easy access is still required.

> *598 I guess to summarize multiple of my previous comments here, I'm just not sure it can be considered unique to exclude areas where your mapping methods does poorly.*

As mentioned before. We are not cherry- picking good values- there is a crucial difference between filtering out poorly calculated values and being aware of the areas in which photogrammetric methods often struggle and masking them out in a structured way.

> *693 Earlier in the paper it was CHF, be consistent*

We will change the currency to CHF in the revised version.

> *697 (GCPs, distribution is not important) - > this statement is not supported by the presented results*

Table 2 shows that the distribution of GCPs is not crucial, since already 1 GCP is sufficient for the correct alignment.

**References**

Bühler, Y., Marty, M., Egli, L., Veitinger, J., Jonas, T., Thee, P. and Ginzler, C.: Snow depth mapping in high-alpine catchments using digital photogrammetry, The Cryosphere, 9, 229–243, doi: 10.5194/tc-9-229-2015, 2015.

Bühler, Y., Adams, M. S., Bösch, R., and Stoffel, A.: Mapping snow depth in alpine terrain with unmanned aerial systems (UASs): potential and limitations, The Cryosphere, 10, 1075-1088, 10.5194/tc-10-1075-2016, 2016.

Cline, D. W.: Digital Photogrammetric Determination Of Alpine Snowpack Distribution For Hydrologic Modeling, Proceedings of the Western Snow Conference,, Colorado State University, CO, USA, 1994.

Deschamps-Berger, C., Gascoin, S., Berthier, E., Deems, J., Gutmann, E., Dehecq, A., Shean, D. and Dumont, M.: Snow depth mapping from stereo satellite imagery in mountainous terrain: evaluation using airborne laser-scanning data, The Cryosphere, 14, 2925–2940, doi: 10.5194/tc-14-2925-2020, 2020.

Deems, J. and Painter, T.: LiDAR Measurement of Snow Depth: Accuracy and Error Sources, in: inproceedings, Proceedings of the 2006 International Snow Science Workshop, Telluride, Colorado, 2006.

Eberhard, L. A., Sirguey, P., Miller, A., Marty, M., Schindler, K., Stoffel, A. and Bühler, Y.: Intercomparison of photogrammetric platforms for spatially continuous snow depth mapping, The Cryosphere, 15, 69–94, doi: 10.5194/tc-15-69-2021, 2021.

Harder, P., Schirmer, M., Pomeroy, J. and Helgason, W.: Accuracy of snow depth estimation in mountain and prairie environments by an unmanned aerial vehicle, The Cryosphere, 10, 2559–2571, doi: 10.5194/tc-10-2559-2016, 2016.

Höhle, J. and Höhle, M.: Accuracy assessment of digital elevation models by means of robust statistical methods, ISPRS830 Journal of Photogrammetry and Remote Sensing, 64, 398–406, doi: 10.1016/j.isprsjprs.2009.02.003, 2009.

Marti, R., Gascoin, S., Berthier, E., Pinel, M. de, Houet, T. and Laffly, D.: Mapping snow depth in open alpine terrain from stereo satellite imagery, The Cryosphere, 10, 1361–1380, doi: 10.5194/tc-10-1361-2016, 2016.

Meyer, J., Skiles, M., Deems, J., Boremann, K. and Shean, D.: Mapping snow depth and volume at the alpine watershed scale from aerial imagery using Structure from Motion, 2021.

Meyer, J. and Skiles, S. McKenzie: Assessing the Ability of Structure From Motion to Map High-Resolution Snow Surface Elevations in Complex Terrain: A Case Study From Senator Beck Basin, CO, Water Resour. Res., 55, 6596–6605, doi: 10.1029/2018WR024518, 2019.

Michele, C. de, Avanzi, F., Passoni, D., Barzaghi, R., Pinto, L., Dosso, P., Ghezzi, A., Gianatti, R. and Della Vedova, G.: Using a fixed-wing UAS to map snow depth distribution: an evaluation at peak accumulation, The Cryosphere, 10, 511–522, doi: 10.5194/tc-10-511-2016, 2016.

Nolan, M., Larsen, C. and Sturm, M.: Mapping snow depth from manned aircraft on landscape scales at centimetre resolution using structure-from-motion photogrammetry, The Cryosphere, 9, 1445–1463, doi: 10.5194/tc-9-1445-2015,2015.

Painter, T. H., Berisford, D. F., Boardman, J. W., Bormann, K. J., Deems, J. S., Gehrke, F., Hedrick, A., Joyce, M., Laidlaw, R., Marks, D., Mattmann, C., McGurk, B., Ramirez, P., Richardson, M., Skiles, S. McKenzie, Seidel, F. C. and Winstral, A.: The Airborne Snow Observatory: Fusion of scanning lidar, imaging spectrometer, and physically-based modeling for mapping snow water equivalent and snow albedo, Remote Sensing of Environment, 184, 139–152, doi: 10.1016/j.rse.2016.06.018, 2016.

Smith, F., Cooper, C., and Chapman, E.: Measuring Snow Depths by Aerial Photography, Proceedings of the Western Snow Conference, Boise, Idaho, USA.1967.

Vander Jagt, B., Lucieer, A., Wallace, L., Turner, D., and Durand, M.: Snow Depth Retrieval with UAS Using Photogrammetric Techniques, Geosciences, 5, 264-285, 10.3390/geosciences5030264, 2015.

---

## Author Response (AR2)

Dear anonymous reviewer. Thank you for the detailed and helpful review.

In the following we will answer the specific comments (the numbers correspond to the lines in our manuscript). The intended arguments are your comments and our response follows:

> *L43 ff: Regarding SWE, it should be mentioned more clearly that the conversion from snow depth to SWE is not trivial and needs additional density assumptions / simulations. In the first paragraph of the current version (p. 2, l. 43ff), the reader might get the impression that you can just use the acquired snow depth maps to derive SWE.*

We will clarify that the calculation of the SWE is not trivial and needs additional density assumptions.

> *In l. 50, you state that snow depth maps are used for validating models. I fully agree. However, using models such as Alpine3D in combination with precipitation scaling methods (Vögeli et al., 2016) cannot be classified as validation. Here, snow depth maps serve as an input to derive simulated snow cover patterns and are used for precipitation scaling. In addition, you should mention that snow depth maps can also be used for snowpack model assimilations as various studies already presented (e.g., Alonso-González et al., 2022).*

You are right that it sounds like Vögeli et al. (2016) used snow depth maps as validation instead as precipitation scaling. We will rephrase this section to clarify and differentiate the various applications.

> *Snow depth information on slopes is an interesting information for ski resorts. Spandre et al. (2017) as well as Ebner et al. (2021; for even more resorts, also in Switzerland) used GNSS techniques mounted snow groomers. In ski resorts, snow depth is derived with the latter method on an almost daily basis; one snow map per season is of course not sufficient for a ski resort and as your maps are acquired around peak SWE, this would rather be at the end of the skiing season and of less importance for the ski resort. However, I would recommend to mention your snow depth maps more in the context of a possible validation/comparison for GNSS derived snow depths on slopes.*

Thanks for the correct hints. However, our snow depth maps have the advantage that they can also serve as reference outside the slopes, for example in ungroomed ski routes. We will modify this section while focusing on our snow depth maps as validation for GNSS derived snow depths as well as reference outside the groomed slopes.

> *The costs of airplane-based photogrammetry is as you mention a bit more economic as ALS, however, I agree with reviewer 1, that both techniques are expensive and not applicable for most regions and (scientific) applications so far. As there is not an order of magnitude of a difference in the price (30.000 – 60.000 CHF vs. 50.000 – 80.000 CHF), I recommend not pronouncing too much that your method is more economic.*

You are right that airplane-based photogrammetry is still very expensive, but taking into account that we planned several flights, the lower costs were crucial for the decision of the platform used. But due to the mentioned concerns by both reviewers we will modify this sentence that airplane-based photogrammetry is only slightly more economic.

*There is a bit of confusion regarding the amount of reference points in the manuscript for the year 2020. In Table 1, 38 reference points are listed, in the text (e.g., Sections 3.2.3 and 4.1) you mention 40 reference points, and in Figure A1 less than 38 or 40 points are shown. Please clarify and be consistent. In addition: Are some reference points used in 2018, 2020 and 2021 the same?*

Thank you for the hint. We measured 37 reference points simultaneously to the Ultracam flight in 2020. One of these points was not suitable because of problems with the GNSS. 2 further points were hard to identify on the images and accordingly also excluded from the workflow. Four of these reference points were used as ground control points. The other 30 points were used as check points. As described in chapter 3.2.3 (Manual reference points), we also measured 10 points in retrospect. 4 of these 10 points were used as ground control points in 2017. In 2018, 3 of these 10 points were applied as ground control points. The other 7 points served as check points. In 2019, 2 points were used as ground control points, 6 points as check points. In 2020, none of these 10 points were included in the workflow. In 2021, 8 points served as check points. However, the selected

We will clarify the number of reference points in a consistent way.

*L. 263: What is meant with outlying areas?*

We excluded the peripheral areas close to the edge of each processed snow depth map because in these areas the number of overlapped images was limited and accordingly also the reliability of the snow depth values decreased. We will remove this detail in the revised version.

*L. 271: I guess there is a typo regarding the NDSI threshold. I assume it should be 0.4 (instead of 0).*

No, 0.0 is the threshold used in our snow depth maps. We tried different values and this threshold led to the best results. In the snow depth map 2022, which is not part of this publication, we modified this threshold to –0.02 because accumulated sahara sand on the snow surface led to different reflectance characteristics.

*Section 4.2.2: I see the point that you increased the upper limit for the 2019 snow depth map generation to 15 m. However, please also point out that this could also lead to a potential offset compared to the other years (e.g., it could lead to a higher potential of including more high value errors (up to 15 m) than in the other years (up to 10 m)).*

We checked the location of snow depth values between 10 and 15 m and they mainly occur in avalanche deposits, which is the reason why we adapted the threshold. Additionally, they also falsely occur in isolated pixels in extremely steep faces. However, we do not think that the adjustment resulted into a significant offset. Therefore, we will add that the modification of the upper threshold does not lead to a significant offset of the statistics.

*Section 4.2.6: Here you present the masking overview for the snow depth map 2020. How is the masking overview for the larger and smaller study areas of the years 2017 and 2018?*

The masking overview (area and relative part) of all years is presented in the appendix B (Table A2). Would you prefer a presentation and description of the overview in the text? Due to the length of the paper, we prefer to limit this overview to 2020. Also, except the outlier mask we do not see an added value of a detailed description for each year.

*Table 6: Besides showing the average and standard deviation, it would make sense to include the median as well as upper and lower quantile values (e.g., Q5 and Q95) for each year.*

We will calculate the median, the Q10 and the Q90 because Q5 and Q95 do not have an added value from our point of view.

> *Figure 12: The 2017 normalized snow depth map shows much darker red areas (right side of image), which are not represented in the colour bar. Are these holes in the map and the underlying upper left image shimmers through with dark? In this case, please let the holes just white.*

Exactly, there are holes in the snow depth maps and additionally the dark Hillshade shimmers through. We will fill the holes with a white layer.

> *In addition to Figure 12, it would be very interesting to show difference maps for four years taking one year as reference (e.g., 2021-2017, 2021-2018, 2021-2019, 2021-2020), then potential offsets / differences would become more visible.*

The length of the paper and the number of figures is already at the limit. But we will add a figure calculating the actual difference of the snow depth maps 2019 and 2020, because 2019 was very snowy and 2020 was slightly below average.

> *Section 6.2.4: As stated above, Vögeli et al. (2016) does not use snow depth maps for validation. I would recommend renaming the title of the Section 6.2.4, e.g., to: 'Validation and Snowpack modelling approaches.*

We will rephrase this section that our data can be used to improve input parameters and as validation for numerous projects in snow-hydrological modelling. We will rename the section corresponding to your suggestion.

> *L.692ff: As mentioned before, it has to be pointed out more clearly that the conversion from snow depth to SWE is not trivial / straightforward.*

We will clarify that the conversion from snow depth to SWE is not trivial in the revised version.